# Global hourly, 5 km, all-sky land surface temperature data from 2011 to 2021 based on integrating geostationary and polar-orbiting satellite data

Aolin Jia[1], Shunlin Liang[2], Dongdong Wang[1], Lei Ma[1], Zhihao Wang[1], Shuo Xu[1]

[1]Department of Geographical Sciences, University of Maryland, College Park, MD, 20742, USA
[2]Department of Geography, University of Hong Kong, Hong Kong, 999077, China

*Correspondence to*: Shunlin Liang (shunlin@hku.hk)

**Abstract.** Land surface temperature (LST) plays a dominant role in the surface energy budget (SEB) and hydrological cycling. Thermal infrared (TIR) remote sensing is the primary method of estimating LST globally. However, cloud cover
leaves numerous data gaps in satellite LST products, which seriously restricts their applications. Efforts have been made to produce gap-free LST products from polar-orbiting satellites (e.g., Terra and Aqua); however, satellite data from limited overpasses are not suitable for characterizing the diurnal temperature cycle (DTC), which is directly related to heat waves, plant water stress, and soil moisture. Considering the high temporal variability of LST and the importance of the DTC, we refined the SEB-based cloudy-sky LST recovery method by improving its feasibility and efficiency and produced a global
hourly, 5 km, all-sky land surface temperature (GHA-LST) dataset from 2011 to 2021. The GHA-LST product was generated using TIR LST products from geostationary and polar-orbiting satellite data from the Copernicus Global Land Service (CGLS) and Moderate Resolution Imaging Spectroradiometer (MODIS). Based on the ground measurements at the 201 global sites from the Surface Radiation Budget (SURFRAD), Baseline Surface Radiation Network (BSRN), Fluxnet, AmeriFlux, Heihe River Basin (HRB), and Tibet Plateau (TP) networks, the overall root mean square error (RMSE) of the
hourly GHA-LST product was 3.31 K, with a bias of -0.57 K and $R^2$ of 0.95. Thus, this product was more accurate than the clear-sky CGLS and MODIS MYD21C1 LST samples. The RMSE value of the daily mean LST was 1.76 K. Validation results at individual sites indicate that the GHA-LST dataset has relatively larger RMSEs for high-elevation regions, which can be attributed to high surface heterogeneity and input data uncertainty. Temporal and spatial analyses suggested that GHA-LST has satisfactory spatiotemporal continuity and reasonable variation and matches the reference data well at hourly
and daily scales. Furthermore, the regional comparison of GHA-LST with other gap-free hourly datasets (ERA5 and Global Land Data Assimilation System, GLDAS) demonstrated that GHA-LST can provide more spatial texture information. The monthly anomaly analysis suggests that GHA-LST couples well with global surface air temperature datasets and other LST datasets at daily mean and minimum temperature scales, whereas the maximum temperature and diurnal temperature range of LST and air temperature (AT) have different anomalous magnitudes. The GHA-LST dataset is the first global gap-free
LST dataset at an hourly, 5 km scale with high accuracy, and it can be used to estimate global evapotranspiration, monitor

extreme weather, and advance meteorological forecasting models. GHA-LST is freely available at https://doi.org/10.5281/zenodo.7487284 (Jia et al., 2022c) and glass.umd.edu/allsky_LST/GHA-LST.

## 1 Introduction

Land surface temperature (LST) is an essential component of the surface radiation budget and a dominant driving force in atmospheric cycling and hydrological balance (Li et al., 2022b; Li et al., 2013b). LST directly reflects the thermal feedback of various land covers towards incoming solar radiation and atmospheric longwave radiation (Liang et al., 2019), and it is employed as an important variable for urban heat island analysis (Liu et al., 2022), permafrost mapping (Zou et al., 2017), and hazard forecasting (Bhardwaj et al., 2017; Mudele et al., 2020; Quintano et al., 2015). Therefore, LST has been extensively utilized as a vital indicator for characterizing regional and global climate change (Zhou et al., 2012; Jin, 2004; Peng et al., 2014). This parameter can be obtained by ground measurements, model simulations, and remote sensing retrievals. However, given the high spatiotemporal heterogeneity caused by various land covers, soil types, topographies, and meteorological conditions (Zhan et al., 2013; Liu et al., 2006; Ma et al., 2021), remote sensing has become the only feasible solution for monitoring LST globally.

LST can be retrieved using thermal infrared (TIR) observations from both polar-orbiting (Wan, 2008; Hulley and Hook, 2009) and geostationary (GEO) satellites (Yu et al., 2008; Freitas et al., 2009). In comparison, GEO satellites provide sub-hourly observations; thus, they can precisely capture the diurnal temperature cycles (DTCs) of the land surface. DTCs characterize the strong temporal variability of LST in a day, which is an important surface thermal property that responds to local environmental changes (Hansen et al., 1995; Sun et al., 2006). Studies have suggested that DTCs are directly related to plant water stress and soil drought (Fensholt et al., 2011; Stisen et al., 2008; Hernandez-Barrera et al., 2017); thus, such a relationship has been utilized for mapping evapotranspiration (Anderson et al., 2011) and soil moisture (Piles et al., 2016). In addition, it helps improve meteorological forecasting through data assimilation (Orth et al., 2017), extreme heat wave assessments (Hrisko et al., 2020; Jiang et al., 2015), crop yield estimations (Anderson et al., 2016), LST spatiotemporal scale conversions (Hu et al., 2020), orbit drift corrections of advanced very high-resolution radiometer (AVHRR) LST data (Jin and Treadon, 2003), and vegetation phenology analyses (Piao et al., 2015). Considering the great potential of DTCs in scientific applications and the high temporal variability of LSTs, accurate diurnal LST datasets are crucial for the research community and public (Chang et al., 2021; Hrisko et al., 2020; Pinker et al., 2019).

TIR sensors on board GEO satellites, such as the Geostationary Operational Environmental Satellites (GOES)-R Advanced Baseline Imager (ABI) and the Meteosat Second Generation (MSG) Spinning Enhanced Visible and InfraRed Imager (SEVIRI), provide exceptional opportunities to record DTCs. However, two inevitable flaws occur when using GEO satellites to observe diurnal LST variations globally, namely, data gaps caused by cloud cover and limited spatial view fields

of individual GEO satellites, which seriously limit the availability of hourly all-sky LST datasets at the global scale. Methods of recovering hourly LSTs have been developed and comprehensively reviewed by Jia et al. (2022a). Currently available gap-free satellite-derived LST products are summarized in Table 1. Some gap-free LST datasets are not listed in the table, such as skin temperature from reanalysis datasets (e.g., ERA5 and MERRA2) (Muñoz-Sabater et al., 2021; Molod et al., 2015), and the results of Coccia et al. (2015) as they assumed that the surface broadband emissivity was equal to 1.

**Table 1: Summary of publicly available gap-free LST products.**

| Study | Spatial coverage | Temporal coverage | Spatial resolution | Temporal resolution | Methodology | Access |
|---|---|---|---|---|---|---|
| Zhang et al. (2022) | global land | 2003 – 2020 | 1 km | 13:30 and 01:30 Local Time (LT) | spatiotemporal interpolation | https://doi.org/10.25380/iastate.c.5078492 |
| Hong et al. (2022) | global land | 2003 – 2019 | 0.5° | daily mean | annual temperature cycle (ATC) model and DTC model | https://doi.org/10.5281/zenodo.6287052 |
| Yu et al. (2022) | global land | 2002 – 2020 | 0.05° | 01:30, 10:30, 13:30, and 22:30 LT | fusion of TIR and reanalysis LSTs | https://www.tpdc.ac.cn/en/data/8e975056-a612-4a4f-b7ef-772720145bf0/ |
| Rains et al. (2022) | Europe | 2018 – 2019 | 1 km | daily | fusion of polar-orbiting and SEVIRI LST data | https://doi.org/10.5281/zenodo.7026612 |
| Jia et al. (2022a) | United States and Mexico | 2017 – 2021 | 2 km | hourly | data fusion and surface energy balance (SEB) correction | http://glass.umd.edu/allsky_LST/ABI/ |
| Xu and Cheng (2021) | China | 2002 – 2020 | 1 km | 13:30 and 01:30 LT | data fusion of TIR and PMW LSTs | https://www.tpdc.ac.cn/en/data/7e5333df-0208-4c4e-ae7e-16dcd29e4aa7/ |
| Zhang et al. (2021) | mainland China | 2000 – 2021 | 1 km | 13:30 and 01:30 LT | data fusion of TIR and reanalysis LSTs | https://www.tpdc.ac.cn/en/data/20aa5af9-a4f8-49e5-9aa6-3a753b4c7815/ |
| Shiff et al. (2021) | global land | 2002 – 2019 | 1 km | 13:30 and 01:30 LT, and daily mean | fusion of TIR LST and modeled temperature | https://zenodo.org/record/3952604#.YnhyrOjMIuU |
| Hong et al. (2021) | global land | 2003 – 2019 | 1 km | daily mean | ATC model and DTC model | http://www.nesdc.org.cn/sdo/detail?id=60f4e35e7e28173cf0c8a771 |
| Li et al. (2021) | United States | 2000 – 2015 | 1 km | 01:30, 10:30, 13:30, and 22:30 LT | data fusion of MODIS TIR and reanalysis LSTs using random forest | http://glass.umd.edu/us_allsky_lst_1km.html |
| Zhao et al. (2020) | China | 2003 – 2017 | 0.05° | monthly | geographically weighted interpolation and fusion with ground measurement | https://doi.org/10.5281/zenodo.3528024 |
| Yan et al. (2020) | North America | 2002 – 2018 | 0.05° | monthly | elevation-based interpolation | https://doi.org/10.5281/zenodo.4184160 |
| Zhang et al. (2019) | Tibet Plateau | 2000 – 2021 | 1 km | 13:30 and 01:30 LT | data fusion of TIR and reanalysis LSTs | https://www.tpdc.ac.cn/en/data/76006ce7-b8dc-4add-bbb5-93f36f4bd26c/ |
| Martins et al. (2019) | Europe and Africa | 2021 – now | 3 km | 30 min | SEB-constrained optimization method | https://landsaf.ipma.pt/en/products/land-surface- |

| | | | | | | temperature/mlstas/ |
|---|---|---|---|---|---|---|
| Duan et al. (2017) | China | 2002 – 2011 | 1 km | 13:30 LT | data fusion of TIR and PMW LSTs | http://www.geodata.cn/data/data details.html?dataguid=398357& docId=1934 |
| Chen et al. (2017) | global land | 2000 – 2020 | 1 km | monthly | mean of daytime and nighttime clear-sky LSTs | https://www.tpdc.ac.cn/en/data/8 caced1f-e10b-41a9-87e1-5001ab432844/ |
| Metz et al. (2017) | global land | 2003 – 2016 | 1 km | monthly | spatiotemporal interpolation | https://doi.org/10.5281/zenodo.1 115666 |
| André et al. (2015) | latitude > 45 ° N | 2000 – 2011 | 25 km | daily | empirical retrieval from PMW BT | https://doi.pangaea.de/10.1594/P ANGAEA.833409 |
| Metz et al. (2014) | Europe | 2000 – now | 250 m | 01:30, 10:30, 13:30, and 22:30 LT | spatiotemporal interpolation and regression-based downscaling | https://www.geodati.fmach.it/eu rolst.html |
| Boukabara et al. (2011) | global land | 2014 – now | 0.09 – 0.5° | 13:30 and 01:30 LT | iterative physical inversion from PMW observations | https://www.avl.class.noaa.gov/s aa/products/search?datatype_fa mily=JPSS_SND |

Table 1 reveals that only a few hourly all-sky LST datasets are currently available, thus an all-sky hourly LST dataset at the global scale is urgently required. In Table 1, the products were divided into three categories based on their associated methodology: data fusion, mathematical interpolation, and cloudy-sky LST estimation based on the surface energy balance (SEB) theory.

Land surface models and reanalysis datasets release simulated hourly skin temperatures continuously, which have been fused with satellite-retrieved LSTs to generate gap-free LSTs (Dumitrescu et al., 2020; Long et al., 2020; Marullo et al., 2014; Ma et al., 2022; Muñoz-Sabater et al., 2021); however, the accuracy of the recovered LSTs is highly dependent on simulation accuracy, especially for continuous cloudy days. In addition, passive microwave (PMW) observations can penetrate clouds

and estimate LSTs in all-sky conditions (Zhang et al., 2019; Wu et al., 2022), and studies have explored fusing such data with TIR LSTs from sensors of polar-orbiting satellites (Zhang et al., 2020; Xu and Cheng, 2021). However, PMW data have limited passing times in a day; thus, they cannot match well with GEO observations. Mathematical interpolation is a popular method of reconstructing hourly LST because an ideal DTC can be parameterized by a harmonic function in the daytime and an exponential function in the nighttime (Duan et al., 2012). However, such parameterization requires at least four

observations per day. Researchers have also tried to improve the feasibility of obtaining gap-free LSTs by combining DTC models with spatial interpolation (Liu et al., 2017) or utilizing a convolutional neural network (CNN) to predict missing values from neighboring clear-sky pixels and texture information (Wu et al., 2019). However, interpolating adjacent clear-sky samples can only obtain theoretical 'clear-sky' LSTs because actual LSTs under clouds are impacted by frequent meteorological changes and cloud cooling/warming effects in the daytime/nighttime (Jin, 2000; Jia et al., 2020).

In comparison, cloudy-sky LST estimates based on SEB exhibit advantages in generating all-sky diurnal LST products at large scales. SEB-based methods include two steps. The first step is to reconstruct theoretical 'clear-sky' LST values for cloudy time points, and the second step is to superpose the cloud effect based on the SEB equation (Jin and Dickinson, 2000; Lu et al., 2011). However, traditional SEB-based methods have limited feasibility because of the high input requirements,

and they can only be used during the daytime. Therefore, Jia et al. (2021) considerably improved upon these methods by incorporating modeling data into the process, and the improved methods can be applied to larger spatial scales. ERA5 surface longwave radiation data were used to build a spatiotemporally evolving model, and the clear-sky GEO LSTs were assimilated to the evolving model to correct its predictions on cloudy days. Moreover, an optimization method was used to compute the cloud effect during both the daytime and nighttime, and complete DTCs could be recovered for hourly LSTs

(Jia et al., 2022a).

The latest SEB scheme has produced all-sky hourly LSTs over the contiguous US (CONUS) and Mexico from ABI data; however, it has a relatively lower computational efficiency owing to the spatiotemporal assimilation framework, which is not easy to apply globally. Furthermore, single GEO data points have a limited spatial view field, which can be solved by

105 combining data from multiple GEO satellites at mid and low latitudes and polar-orbiting satellites (Terra + Aqua) at high latitudes. Two polar-orbiting satellites pass high latitudes subhourly and provide frequent observations as GEO satellites. This strategy has been successfully utilized to generate global hourly Clouds and Earth's Radiant Energy System (CERES) radiation products (Loeb et al., 2018); however, few studies have focused on estimating all-sky LSTs by combining polar-orbiting and GEO satellites.

In this study, we produced a global, hourly, all-sky LST dataset (GHA-LST) from 2011 to 2021 at a 5 km scale, and a comprehensive assessment was implemented using 201 ground sites worldwide. Global clear-sky LSTs were obtained by combining GEO LSTs from the Copernicus Global Land Service (CGLS) and Moderate Resolution Imaging Spectroradiometer (MODIS) MxD21 LST swath products, and a more efficient spatiotemporal assimilation scheme was

115 proposed. It represents the first available global all-sky LST scheme on an hourly time scale with satisfactory accuracy based on global site validation; thus, it has great potential for use in analyzing global thermal dynamics, atmospheric cycling, and hydrological budgets.

## 2 Data and Method

### 2.1 Data

The proposed GHA-LST dataset was recovered from a combination of clear-sky LST products, including the CGLS hourly LSTs, which cover mid and low latitudes, and MOD/MYD21 instantaneous swath LSTs, which cover high latitudes. ERA5 provides dynamic surface temperature signals for building a temperature-time-evolving model, and CERES global hourly

surface radiation products were used to compute the cloud effect. In addition, the all-sky LST data were comprehensively assessed based on globally distributed sites collected from the Surface Radiation Budget (SURFRAD), Baseline Surface Radiation Network (BSRN), Fluxnet, AmeriFlux, Heihe River Basin (HRB), and Tibetan Plateau (TP) networks.

### 2.1.1 Input data

CGLS LST provides hourly clear-sky LST retrievals from a constellation of GEO satellites, including multiple generations of Meteosat Second Generation (MSG), Multifunctional Transport Satellite (MTSAT)/Himawari, MSG Indian Ocean Data Coverage (IODC), and GOES. The product is released as a global product that covers land surfaces worldwide within the 60° S to 70° N latitudes. The generalized split window (GSW) algorithm and dual algorithm (DA) in mono- and dual-channel forms were used to retrieve LST from top-of-atmosphere (TOA) brightness temperatures (BTs) in thermal infrared window channels (Freitas et al., 2013). Based on the ground validation, the accuracy ranges from 1.83 K to 3.70 K.

MOD/MYD21 swath instantaneous LST products (Hulley et al., 2016) were used to provide LST over the rest of the land surface, which mainly covered high latitudes. A temperature/emissivity separation (TES) algorithm was used to retrieve the LST in the MOD/MYD21 products. It showed comparable accuracy to that of MOD/MYD11 (Wan, 2008) for most land cover types and performed better in bare land regions (Li et al., 2020; Yao et al., 2020). Level 3 MOD/MYD21 LST products provide gridded LST data in sinusoidal projection such that LST images were available four times a day and pixel locations were fixed. This data format is convenient for users; however, many valid retrieval values are lost at higher latitudes due to reprojection. In fact, dozens of times can be recorded by combining two polar-orbiting satellites, which is comparable to GEO observations at mid and low latitudes. Therefore, to fully incorporate the available clear-sky retrievals, MOD/MYD21 swath instantaneous LST data were used in this study. All swath images were converted to the Climate Modeling Grid (CMG) individually and then aggregated to the same spatial resolution as CGLS LST. The instantaneous observations were used for averaging only when they were within a 30-minute window centered at the CGLS recording time (UTC standard time). In addition, to minimize the impact of retrieval uncertainty, records with a view zenith angle greater than 40° were not used in this study (Li et al., 2014; Guillevic et al., 2013). By using this strategy, we can obtain a recording frequency over polar regions that is comparable to that of the CGLS LST data; however, using this strategy does not mean that significantly more clear-sky LST samples will be obtained because cloud cover persists at high latitudes (King et al., 2013). This process only ensures that clear-sky LSTs at high latitudes are included in as many observations as possible.

To obtain continuous surface thermal variational signals, surface upward longwave radiation (ULW) and downward longwave radiation (DLW) simulated by ERA5 were used to build the LST time-evolving model. Satellite-derived broadband emissivity (BBE) was obtained from the Global LAnd Surface Satellite (GLASS) (Liang et al., 2021). We calculated the LST series using ERA5 DLW and ULW data simulated using clear-sky scenarios, which were generated based on real atmospheric and meteorological conditions, although clouds were assumed to be absent. The ERA5 clear-sky

scenario was used because utilizing cloud radiative forcing calculated from global satellite data is more accurate than using the simulated results from the reanalysis (Wang and Dickinson, 2013). The clear-sky LST retrievals were then assimilated into the time-evolving model to obtain continuous LSTs without cloud gaps, and the cloud cooling/warming effect was then estimated and superposed from satellite radiation products. Essentially, in this revised SEB-based recovery method, the temperature change signals under cloud cover were divided into two parts: the evolving model provided the LST variations due to real-time meteorological changes under clouds and satellite radiation products estimated the cloud cooling and warming effects caused by cloud radiative forcing.

Global hourly surface DLW and downward shortwave radiation (DSR) from CERES satellite products were used to estimate the cloud effect. To monitor cloud radiative forcing, the CERES project retrieved global, gap-free, hourly DSR and DLW in both all-sky (realistic) and theoretically cloud-free conditions (Doelling et al., 2016). CERES utilized the same strategy as this study to generate global hourly radiation products by combining remote sensing observations from multiple GEO sensors and two MODIS sensors. The CERES surface shortwave radiation and longwave radiation (Doelling et al., 2013) were estimated based on the Langley Fu–Liou radiative transfer theory (Fu et al., 1997), the cloud properties were obtained from microwave cloud products (Minnis et al., 2020), and the aerosol optical depth was based on the MODIS aerosol product (Remer et al., 2006). Surface CERES downward radiation fluxes have an overall bias (standard deviation) of 3.0 W m$^{-2}$ (5.7%) for shortwave and −4.0 W m$^{-2}$ (2.9%) for longwave radiation, which have been validated based on 85 sites (Rutan et al., 2015). CERES has been extensively evaluated and is generally considered a benchmark for satellite radiation products for assessments and inter-comparisons (Jia et al., 2018; Li et al., 2022a; Wang and Dickinson, 2013).

To improve the production efficiency, the complicated downward longwave parameterization schemes in Jia et al. (2022a) were replaced by directly exploiting the CERES dataset and converting its cloud radiative forcing into the corresponding cloud cooling/warming effect. Specifically, the CERES DSR difference between all-sky and clear-sky schemes was considered cloud DSR forcing, and combined with the GLASS surface albedo data, the cloud net shortwave forcing was computed. Cloud DLR forcing represents the difference between CERES all-sky and clear-sky DLR products, and the corresponding net longwave forcing was estimated using an optimization method (Section 2.2.4). In addition, according to previous studies (Wang and Dickinson, 2013; Zhang et al., 2015), the impact of the coarse spatial resolution of CERES downward radiation can be ignored because it has less heterogeneity than surface variables. CERES products were bilinearly interpolated to match the spatial scale of the CGLS LST. However, this assumption may introduce a certainty degree of uncertainty in areas with rugged terrain because complicated terrain in a coarse pixel may still affect the downward radiation components and increase the heterogeneity.

The GLASS 0.05° land surface albedo and BBE were used for the net radiation calculation (Liang et al., 2021), and the GLASS leaf area index (LAI) was used for computing ground heat flux from the net radiation. All input data were preprocessed using bilinear resampling to match the CGLS LST. The input metadata are listed in Table 2.

**Table 2: Metadata input for production.**

| Product | Variable | Temporal Resolution (°) | Spatial Resolution (°) | Usage |
|---------|----------|-------------------------|------------------------|-------|
| CGLS | clear-sky LST | hourly | 0.045 | LST for recovery |
| Swath MOD/MYD21 | clear-sky LST | instantaneous | 1 km | LST for recovery |
| ERA5 | clear-sky DLW & ULW | hourly | 0.25 | time-evolving model |
| GLASS | BBE | 8-day | 0.05 | time-evolving model |
| CERES | all-sky and clear-sky DSR and DLW | hourly | 1 | cloud effect |
| GLASS | surface albedo | 8-day | 0.05 | cloud effect |
| GLASS | LAI | daily | 0.05 | cloud effect |

**2.1.2 Ground measurement**

To comprehensively assess the accuracy of the proposed GHA-LST dataset, globally distributed in situ sites must be collected for ground validation. We processed the records from SURFRAD, BSRN, Fluxnet, AmeriFlux, HRB, and TP networks. SURFRAD was established in 1993 and consistently provides long-term ground measurements of the surface radiation components over CONUS for climate research and remote sensing retrieval validation (Augustine et al., 2000). The BSRN is a combined network of globally distributed sites from several projects (Driemel et al., 2018), and it provides records with strict data quality maintenance; thus, it is usually used as a reference dataset for radiation product validation at the global scale. Fluxnet includes hundreds of ground sites that have been utilized for global LST validation and analysis (Xing et al., 2021). AmeriFlux measures radiation and carbon fluxes over South and North America (Novick et al., 2018). The HRB network is from the Chinese Heihe Watershed Allied Telemetry Experimental Research (HiWATER) project (Li et al., 2013a), and the TP network is from the Third Tibetan Plateau Atmospheric Scientific Experiment (TIPEX-III) (Zhao et al., 2018), and these datasets have been used for LST validation at the kilometer scale (Xing et al., 2021).

In addition, only raw observations marked as 'good quality' were used for validation. Site-measured ULW and DLW were used to compute LSTs with a GLASS BBE based on the Stefan–Boltzmann law. LST raw records within a 30-minute

window centered at each UTC time period were then aggregated for hourly LST validation. The daily mean LST was further
aggregated as long as 24 hourly LSTs were available in a day, which is sufficient to represent the entire DTC based on Jia et
al. (2022a). The validation period was 2011–2020. The following validation metrics are used in this study: N is the sample
amount; bias, also called mean bias error (MBE), represents the systematic errors/differences between LST products and
ground measurements; root-mean-square-error (RMSE) characterizes the actual uncertainty caused by bias and random error;
and $R^2$ indicates the overall goodness of fit based on a 1:1 line. These metrics are commonly used for LST validation. The
standard deviation (SD) of the differences between LST products and site measurements was not used because it provides
similar information as the RMSE but cannot reflect errors caused by systematic bias; thus, the SD is generally smaller than
the RMSE.

The proposed GHA-LST dataset has a spatial resolution of approximately 5 km, although some sites may not be
representative of the corresponding pixels. To remove sites with higher heterogeneity, we utilized two removal strategies.
The first is based on 30 m LSTs from the United States Geological Survey (USGS) Landsat 8 Level 2 Collection 2, in which
clear-sky Landsat LSTs were extracted from all site locations from 2013 to 2020 and the average 30 m LSTs were extracted
within the corresponding 5 km pixel range; then, the RMSE at each site was computed as the site representativeness using
the 30 m LSTs paired with the averaged 5 km LSTs. One site was marked if it had a considerably larger RMSE, indicating
that there were larger LST differences between the 30 m and 5 km scales. The second strategy considered the MYD21C1
0.05° LST as a benchmark LST product on a 0.05° spatial scale. As MYD21 has been comprehensively validated and
produces results with high and stable accuracy at a global scale (Li et al., 2020; Yao et al., 2020; Hulley, 2015), we argue
that if one site has a significantly larger RMSE in the validation of MYD21C1 samples with good quality, then the site will
have low representativeness at the 5 km spatial scale. Pre-processed sites detected by either of these two strategies were
excluded from this study, and the analysis results are shown in Figure 1. The selection threshold of each strategy was equal
to the average RMSE + 2 × standard deviation for all sites.

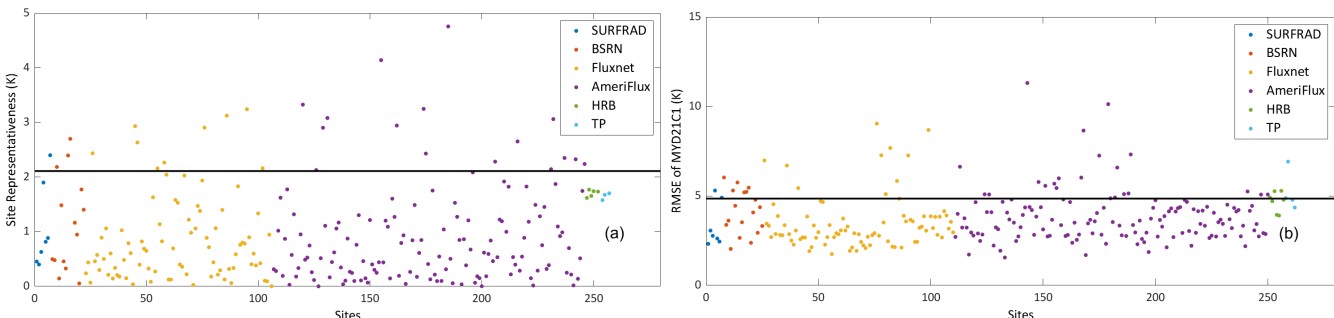

**Figure 1. LST site selection results based on the (a) site representativeness calculated by Landsat samples and (b) site RMSEs of
MYD21C1. The thresholds are the black lines, and sites with RMSEs higher than the line were masked out. Abbreviations:
SURFRAD, Surface Radiation Budget; BSRN, Baseline Surface Radiation Network; HRB, Heihe River Basin (HRB); TP, Tibetan
Plateau.**

Based on Figure 1, 201 global sites were included in this study, including five SURFRAD sites, 11 BSRN sites, 91 Fluxnet sites, 89 AmeriFlux sites, three HRB sites, and two TP sites. The distribution of sites is shown in Figure 2. Additionally, as site selection influences the final validation statistics, we also validated samples from clear-sky MYD21C1 and CGLS for accuracy comparisons, and the accuracy level of the validation results can be used as the reference for GHA-LST.

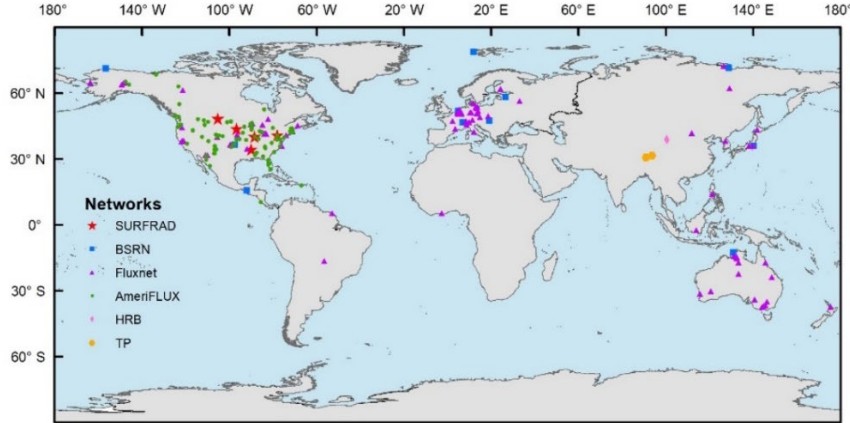

**Figure 2. Global distribution of the utilized 201 LST sites. Abbreviations: SURFRAD, Surface Radiation Budget; BSRN, Baseline Surface Radiation Network; HRB, Heihe River Basin (HRB); TP, Tibetan Plateau.**

## 2.2 Methods

### 2.2.1 Production framework

Jia et al. (2022a) developed a three-step framework to generate all-sky hourly LSTs on a regional scale. In this study, we revised this framework to improve its efficiency and feasibility worldwide. A flowchart of the framework is shown in Figure 3.

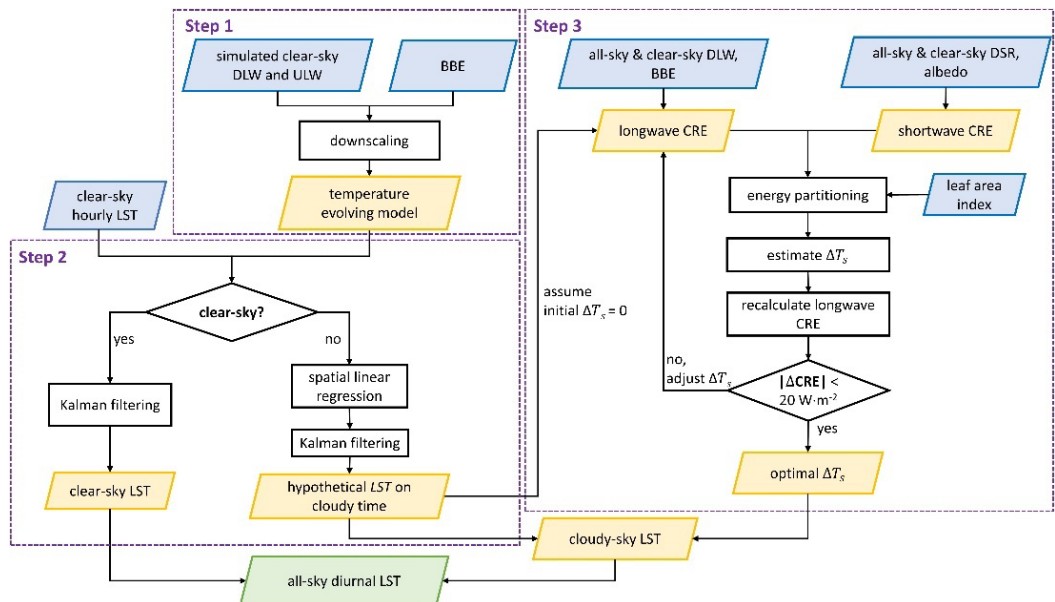

**Figure 3. Flowchart of the proposed GHA-LST dataset production. Abbreviations: DLW, downward longwave radiation; ULW, upward longwave radiation; DSR, downward shortwave radiation; BBE, broadband emissivity; LST, land surface temperature; CRE, cloud radiative effect.**

In the first step, a time-evolving model of clear-sky LSTs was designed based on the ERA5 LST series at each pixel location (Sect. 2.2.2). The ERA5 LST series was computed from the DLW and ULW from the clear-sky simulation scenarios, and it provides continuous variational information on LST without considering the cloud cooling/warming effect. Such variational information under clouds can be attributed to advective meteorological changes and air movement. The ERA5 skin

temperature was not involved because we calculated the cloud effect based on satellite-derived radiation products, which are more reliable at a global scale.

In the second step, the Kalman filter (KF) was used to assimilate available clear-sky LST retrievals into the time-evolving model to correct the predictions for times with cloud cover and then theoretical 'clear-sky' LSTs were reconstructed. In the

original framework of Jia et al. (2022a), three-dimensional data assimilation was utilized to generate a spatiotemporally dynamic model; however, this process is time-consuming, particularly when working at a large spatial scale. We replaced the spatial module in the assimilation by linear regression, which still works well to incorporate spatially adjacent clear-sky retrievals (Sect. 2.2.3). After this step, hypothetical LSTs were reconstructed during times with cloud cover. The cloud effect was further superimposed in the final step.

In the third step, the cloud effect was estimated from the satellite radiation products (Section 2.2.4). In the original framework, cloud longwave radiative forcing was computed based on a series of parameterization schemes; however, the

scheme was not well assessed at the global scale. Therefore, to simplify the calculation and improve the feasibility filling gaps over a large spatial scale, we replaced the DLW parameterization with CERES clear-sky and all-sky DLW products that have been assessed globally (Wang and Dickinson, 2013). The cloud effects at daytime and nighttime were determined by searching for the optimal cloud radiative effect (CRE) values to meet the SEB. The final clear-sky LSTs are the assimilated results at the clear-sky time, and the cloudy-sky LSTs are the reconstructed LSTs from the second step plus the optimal cloud effect.

### 2.2.2 Time-evolving model

A time-evolving model describes how LSTs change at a certain pixel over time, and it characterizes relative variation based on the ERA5 LST rather than absolute magnitudes. The ERA5 LST series was initially downscaled to match the CGLS LST using elevation (Duan et al., 2017). The evolving model can be mathematically represented by Eq. 1-2:

$$LST_{t,d} = F_{t,d} \times LST_{t,d-1}, (1)$$

$$F_{t,d} = (1 + \frac{1}{Z_{t,d}+\delta})\frac{dZ_{t,d}}{dt}, (2)$$

where $LST_{t,d}$ is the LST predicted by the model on day $d$ at hour $t$ and $F_{t,d}$ is the prediction operator, which is generated based on the temperature temporal profile $Z_{t,d}$ (temperature difference between $d$ and $d-1$ at hour $t$, i.e., the difference with the LST 24 h before), and $\delta = 0.01$ avoids a null denominator. The model evolves from day to day for each hour of the day (HOD) because the modeling bias is self-correlated at the same HOD on different days (Marullo et al., 2014). The correction of the data assimilation can be better inherited based on the evolving structure of Eq. 1-2. In addition, only the difference information was used in the study, which can minimize the impact of the uncertainty of ERA5 LST, especially from its systemic bias (Nogueira et al., 2021).

The time-evolving model provides continuous temperature variation; however, the modeling process must be consistently corrected by assimilating available clear-sky retrievals. In addition, partially cloud-covered yet retrieved ('likely cloud contaminated') pixels were masked out before assimilation. The detection method follows Jia et al. (2022a). One clear-sky retrieval is excluded if it has a substantially larger absolute difference with the modeled LST (three standard deviations) than neighboring days within ±15 days, which assumes that modeled LSTs have fewer anomalies than directly retrieved values. It should be noted that some input data (e.g., CERES and reanalysis data) are not available at near real time (NRT); moreover, this 'likely cloud contamination' detection method also requires a 30-day time window for high-quality clear-sky LST selection, which means that the proposed cloudy-sky LST recovery method cannot be used for NRT all-sky LST production.

### 2.2.3 Kalman filter

The KF was used to assimilate clear-sky LST retrievals to correct the evolving model prediction because real-time retrievals are discontinuous while the evolving model prediction is continuous. When a retrieval value is available, a weighted average

is implemented between the prediction and the observation based on their individual uncertainties, and then the prediction is corrected. When observations are not available, the prediction will be implemented based on the updated prediction from the former step. Therefore, a continuous LST series can be generated using this iterative process. The KF can be mathematically represented as follows:

$$\hat{T}_{t,d}^{\ -} = A_{d-1}^t \hat{T}_{t,d-1} + \omega_{d-1}^t, (3)$$

$$\hat{T}_{t,d} = \hat{T}_{t,d}^{\ -} + K_d^t (T_{t,d} - \hat{T}_{t,d}^{\ -}), (4)$$

$$K_{t,d} = P_{t,d}^{-} (P_{t,d}^{-} + R)^{-1}, (5)$$

$$P_{t,d} = (I - K_{t,d}) P_{t,d}^{-}, (6)$$

where $\hat{T}_{t,d}^{\ -}$ is the temperature prediction at time $t$ on day $d$ from the prediction of $d-1$ and $A_{d-1}^t$ is the prediction process (Sect. 2.2.2) with a prediction error of $\omega_{d-1}^t$. The symbol " $-$ " next to a variable indicates that the variable is an initial prediction without assimilation correction. The modeling prediction is propagated to $P_{t,d}^{-}$ after this prediction.

If an observation ($T_{t,d}$) is available, $\hat{T}_{t,d}^{\ -}$ will be corrected using the Kalman gain $K_d^t$ (Eq. 4), which was determined by the relative magnitude of the squared uncertainty of the model prediction $P_d^{-}$ and the satellite retrieval $R$. $R$ is the squared retrieval uncertainty at each pixel of CGLS and MODIS, and it is calculated based on the 'ERRORBAR_LST' or 'LST_err' information in the files. The initial modeling uncertainty was calculated based on a comparison between the ERA5 and satellite retrievals at clear-sky time in the data series. The assimilation started in October 2010, and the model prediction reached a stable status before the product release date; thus, the initial value did not affect the output. The prediction error of $d$ will also be corrected to $P_{t,d}$ (Eq. 5). The next day will then be iteratively predicted. If there are no observations, then the LST will be automatically predicted on *day d* without correction.

Jia et al. (2022a) used a spatial KF module that can assimilate spatially adjacent clear-sky pixels into the evolving model; however, this process is time-consuming and impractical for global production. In this study, if an observation on *d* was available, then the time-evolving model was corrected by KF normally; otherwise, a 30 × 30 [~150 km, referred to Jia et al. (2022a)] spatial window was set for the time-evolving model, and clear-sky pixels and their corresponding ERA5 LST within the spatial window will regress a linear conversion model, and the missing LST at the center pixel on *d* will be predicted from its corresponding ERA5 LST using such a linear relationship. Essentially, a local linear relationship replaced the spatial KF module, although it still effectively incorporated the adjacent clear-sky retrieval, and the computation efficiency was significantly improved. After the center LST was estimated by linear regression, it was considered the available retrieval for KF correction on *d*. If the spatial window did not have available clear-sky retrievals, then the time-evolving model predicted the LST on *d* based on the results on *d-1*.

## 2.2.4 Cloud effect estimation

After data assimilation, the LSTs under clouds was initially predicted without considering the cloud effect. The cloud effect is the temperature warming/cooling effect caused by changing the SEB from clouds, which can be estimated using cloud radiative forcing. The SEB equation is as follows:

$$R_n = R_s^d(1-\alpha) + \varepsilon R_l^d - \sigma\varepsilon T^4 = G + LE + H, (7)$$

where $R_n$ is the surface net radiation, $R_s^d$ is the DSR, $\alpha$ is the surface albedo, $\varepsilon$ is the surface broadband emissivity (BBE), $R_l^d$ is the DLW, σ is the Stefan–Boltzmann constant, and $T$ is the LST. $R_n$ is partitioned into latent heat ($LE$), sensible heat ($H$) and ground heat ($G$). Cloud coverage changes $R_n$, which is called cloud radiative forcing. By following the land surface analysis (LSA SAF) GEO evapotranspiration product, $G$ can be parameterized as follows:

$$G = R_n \times 0.5exp(-2.13(0.88 - 0.78exp(-0.6LAI))), (8)$$

where the $G$ is set to 0.15 (0.05) $R_n$ for bare land (permanent snow/ice). Based on the conventional force-restore method (Jin and Dickinson, 2000), $G$ can be represented as follows:

$$G = k_g \frac{\partial T}{\Delta Z} = k_g \frac{T-T_d}{\Delta Z}, (9)$$

where $k_g$ represents surface thermal conductivity (W m$^{-1}$ K$^{-1}$) and $\Delta Z$ is the responding surface depth, which is set to 0.1 m. The deep layer temperature ($T_d$) is assumed to have little response towards SEB; thus, Eq. 9 can be rewritten as follows:

$$\frac{\partial G}{\partial T_s} = \frac{\partial}{\partial T_s}\left[k_g \frac{T_s-T_d}{\Delta Z}\right] \approx \frac{k_g}{\Delta Z}, (10)$$

Accordingly, the change in $G$ ($\partial G$) caused by cloud coverage can be directly converted into the variation in LST, and $\partial G$ is determined by partitioned cloud radiative forcing. That is, by knowing any two of the three variables ($\partial G$, $\Delta T_s$, and $k_g$), the other can be estimated. According to Jia et al. (2022a), $k_g$ was predetermined based on a continuous temperature series from the assimilation results and corresponding radiation data:

$$k_g = \Delta h \frac{\overline{G_{noon}}-\overline{G_{sr}}}{\overline{T_{noon}}-\overline{T_{sr}}}, (11)$$

where $\overline{G_{noon}}$ ($\overline{T_{noon}}$) and $\overline{G_{sr}}$ ($\overline{T_{sr}}$) are the monthly averaged ground heat (clear-sky LST) within ±15 days at noon and sunrise, respectively, which are considered because morning warming can mainly be attributed to the SEB. The continuous data series from the data assimilation step ensures sufficient sampling for the $k_g$ calculation. Monthly means were used to minimize the disturbance of daily variation. Then, the cloud radiative forcing needs to be determined to estimate $\Delta T_s$:

$$R_c = (1-\alpha)\left(R_{s,cld}^d - R_{s,clr}^d\right) + \varepsilon\left(R_{l,cld}^d - \sigma T_{cld}^4\right) - \varepsilon\left(R_{l,clr}^d - \sigma T_{clr}^4\right), (12)$$

where $R_c$ is the cloud radiative forcing, $R_{s,cld}^d$ ($R_{s,clr}^d$) is the cloudy-sky (clear-sky) DSR, and $R_{l,cld}^d$ ($R_{l,clr}^d$) is the cloudy-sky (clear-sky) DLW. The shortwave variables, BBE, and DLW can be obtained from the global radiation products.

The cloudy-sky LST ($T_{cld}$) equals the reconstructed LST ($T_{clr}$) plus the cloud effect ($\Delta T_s$), and $T_{clr}$ is reconstructed from step two. $\Delta T_s$ is unknown for the radiative forcing calculation and represents the ultimate target of this step. Therefore, the optimal $\Delta T_s$ must be determined based on the optimization method to satisfy the SEB. Following Jia et al. (2022a), $\Delta T_s$ was initially assumed to be 0 K and the initial $R_c$ was obtained based on Eq. 12. After energy partitioning through the LAI, $G$ is computed and the updated $\Delta T_s$ is estimated using $k_g$; thus, $R_c$ can be recomputed. By iteratively comparing the $R_c$ differences and adjusting $\Delta T_s$ (step = 0.05 K), the surface energy budget is balanced ($|\Delta CRE| < 20$ W·m$^{-2}$, see Figure 3). The threshold of 20 W·m$^{-2}$ is the current accuracy level of the longwave radiation products (Wang et al., 2020).

## 3 Results and discussion

### 3.1 Overall assessment

Based on all paired samples from the proposed GHA-LST dataset and 201 sites from 2011-2020, the overall RMSE of the all-sky GHA-LST is 3.31 K, with a bias of -0.57 K and R$^2$ of 0.95. As site selection may influence the accuracy statistics, the accuracy of GHA-LST was also compared with that of the CGLS and MYD21C1 data (Figure 4). Figure 4a shows the extracted clear-sky samples from the CGLS, whereas the corresponding clear-sky results from GHA-LST are shown in Figure 4b, which has the same sampling amount as Figure 4a. The recovered cloudy-sky LST at the corresponding CGLS cloudy time of Figure 4a was validated in Figure 4c. Such an accuracy comparison between the GHA-LST and CGLS data is sufficiently fair; thus, we also compared the accuracies between GHA-LST and MYD21C1 (Figure 4d-4f). Because MYD21C1 may have a slightly different observation time (< 0.5 h) relative to GHA-LST, it was converted to the nearest UTC o'clock based on the diurnal cycle recorded by site observations to match the GHA-LST recording time.

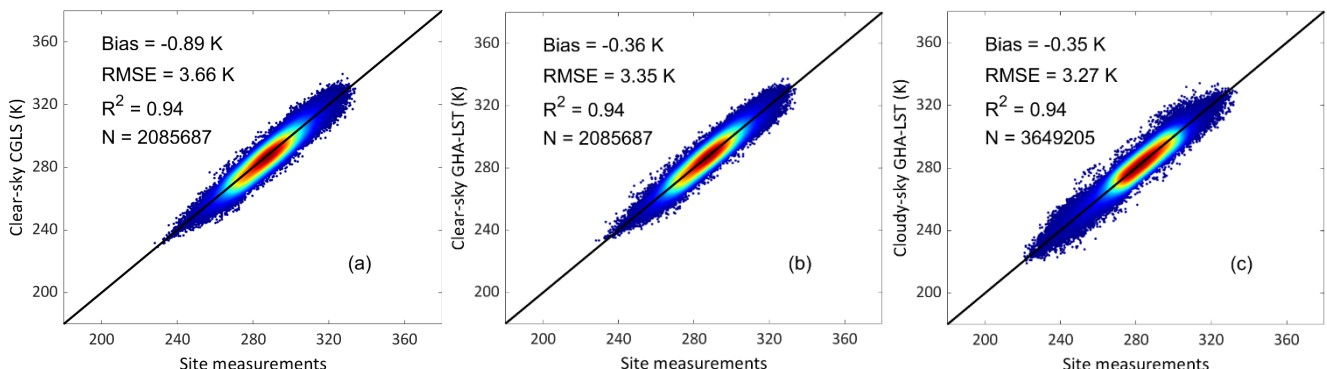

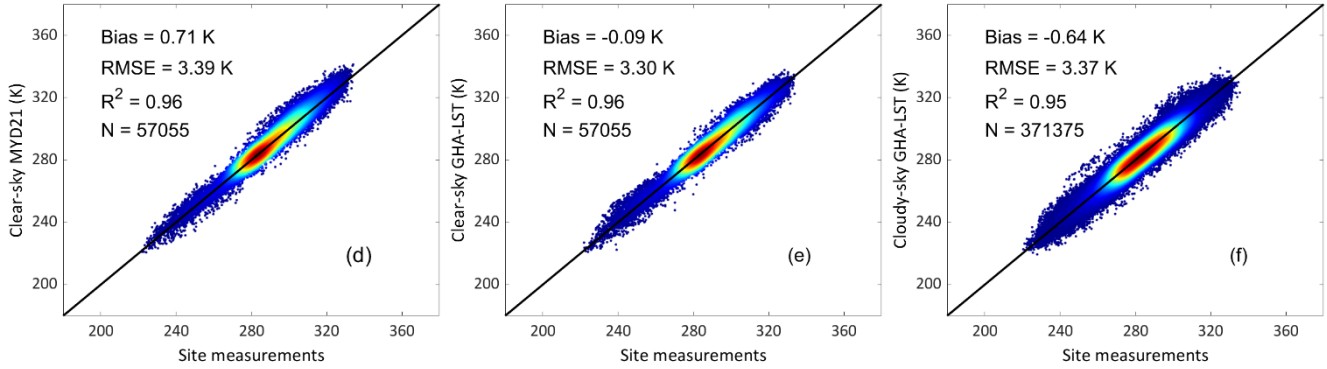

**Figure 4. Density scatterplots of hourly LST samples from (a) CGLS clear-sky retrievals, (b) GHA-LST clear-sky samples corresponding to (a), (c) GHA-LST cloudy-sky samples corresponding to CGLS cloudy time, (d) MYD21C1 clear-sky retrievals, (e) GHA-LST clear-sky samples corresponding to (d), and (f) GHA-LST cloudy-sky samples corresponding to MYD21C1 cloudy time.**

The proposed GHA-LST dataset had better accuracy than the CGLS and MYD21C1 data under both clear-sky and cloudy-sky conditions (Figure 4). Compared to the CGLS (Figure 4a), the clear-sky GHA-LST had a similar RMSE (3.35 K, Figure 4b), which is reasonable because most of the clear-sky GHA-LST samples were derived from clear-sky retrievals from the CGLS. The partially cloud-contaminated samples were marked during production (Section 2.2.2) and considered cloudy pixels. The results of MYD21C1 (Figure 4d) only utilized samples that were marked as 'good quality' and passed a cloud contamination test (Section 3.6 in Ma et al. (2020)). In comparison, GHA-LST produced similar accuracy with high-quality MODIS samples (Figure 4e) and represented a stable accuracy (RMSE = ~3.3 K with few biases) under both clear-sky and cloud-sky conditions. Based on the sampling amount (Figure 4a–c), the number of cloudy pixels was nearly 1.5 times higher than the number of clear-sky pixels, indicating the importance and necessity of the proposed GHA-LST dataset. As MYD21C1 (Figure 4d–f) only includes samples from noon and midnight, it has significantly fewer samples than the CGLS group (Figure 4a–c). Studies on similar topics, such as the recovery of all-sky MSG/SEVIRI LST (Martins et al., 2019), produced RMSEs of 2.1~3.7 K at 3 sites. However, direct comparisons of the validation statistics are difficult because substantially different sites and sampling amounts were utilized in this study.

Figure 5 illustrates that the GHA-LST process has a considerably higher performance in obtaining daily mean LSTs. GHA-LST has a similar RMSE to the daily mean CGLS, with 24 values in a day retrieved by satellites (Figure 5a and b); however, GHA-LST has substantially more available samples than the CGLS clear-sky results. In comparison, the daily mean computed from the average of the paired daytime/nighttime MYD21C1 has the largest RMSE, with a clear bias of 2.41 K. However, MODIS LSTs have been more widely used than GEO LSTs, and many studies have obtained daily mean LSTs by simply averaging two instantaneous Aqua retrievals at noon and midnight (Ouyang et al., 2012; Chen et al., 2017; Zou et al., 2017). This study suggests that the proposed GHA-LST dataset can significantly improve the accuracy and data availability of the daily mean LST.

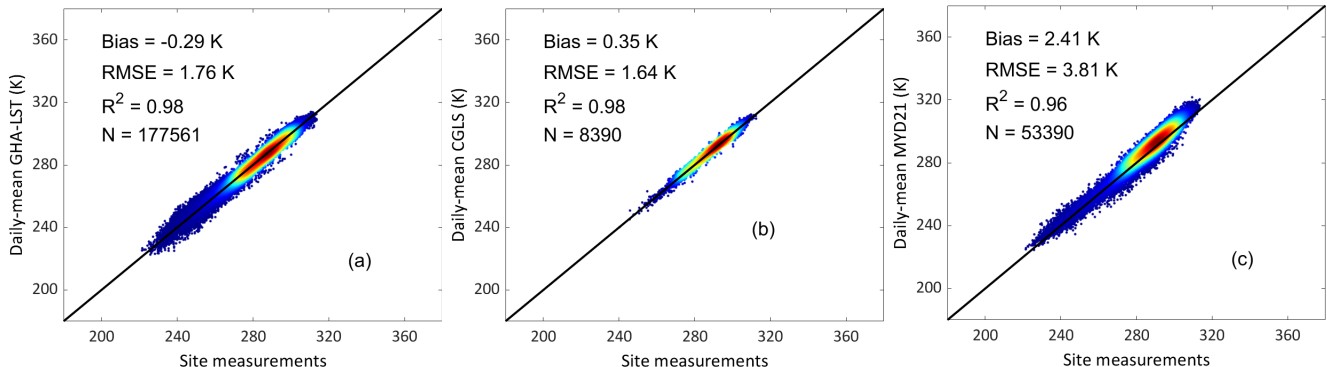

**Figure 5. Density scatterplots of the daily mean LST samples from (a) GHA-LST, (b) CGLS, and (c) average daytime and nighttime MYD21C1 pairs.**

To demonstrate the accuracy and stability of the GHA-LST under different surface conditions, the hourly samples were further differentiated based on land cover types. Land cover data are obtained from the MCD12Q1 International Geosphere-Biosphere Programme (IGBP) classification. The validation statistics are listed in Table 3.

**Table 3: Validation statistics for different land cover types.**

| Land Cover | Clear-sky Samples | | | | Cloudy-sky Samples | | | |
|---|---|---|---|---|---|---|---|---|
| | Bias (K) | RMSE (K) | $R^2$ | N | Bias (K) | RMSE (K) | $R^2$ | N |
| Evergreen needleleaf forests | 0.03 | 3.29 | 0.90 | 187,100 | 0.50 | 2.78 | 0.89 | 281,421 |
| Evergreen broadleaf forests | 0.22 | 3.03 | 0.89 | 37,907 | 1.34 | 2.27 | 0.95 | 39,350 |
| Deciduous broadleaf forests | -1.26 | 3.26 | 0.94 | 190,753 | -0.54 | 2.61 | 0.94 | 275,551 |
| Mixed forests | -1.03 | 2.97 | 0.96 | 90,369 | 0.19 | 2.83 | 0.95 | 176,405 |
| Closed shrublands | 1.19 | 3.80 | 0.94 | 21,781 | 0.61 | 3.21 | 0.91 | 12,760 |
| Open shrublands | -0.49 | 2.87 | 0.97 | 124,904 | -0.28 | 4.07 | 0.96 | 184,840 |
| Woody savannas | -0.20 | 3.26 | 0.93 | 263,142 | 0.34 | 3.63 | 0.94 | 602,743 |
| Savannas | -0.56 | 3.06 | 0.95 | 182,085 | 0.06 | 3.03 | 0.96 | 318,231 |
| Grasslands | -0.25 | 3.66 | 0.84 | 571,722 | 0.19 | 3.95 | 0.92 | 552,178 |
| Permanent wetlands | 1.29 | 3.81 | 0.91 | 11,265 | 0.13 | 3.35 | 0.96 | 72,375 |
| Croplands | -0.28 | 3.10 | 0.95 | 522,715 | 0.15 | 2.72 | 0.94 | 790,227 |
| Urban | -0.61 | 3.80 | 0.94 | 59,698 | -0.44 | 3.11 | 0.92 | 88,839 |
| Barren | -2.01 | 4.02 | 0.98 | 3,925 | -1.95 | 4.59 | 0.94 | 4,812 |

Table 3 indicates that the GHA-LST has stable accuracy under both clear-sky and cloudy-sky conditions for various land cover types. However, some clear biases were found for the forest and barren land cover types, which could be caused by split-window retrieval errors under clear-sky conditions due to the large emissivity uncertainty (Li et al., 2022b). In comparison, high $R^2$ values in these regions reflect the ability of the GHA-LST to capture regional temperature variations.

### 3.2 Individual site validation

Considering that 201 global sites were utilized, the RMSEs at individual sites can reflect the spatial pattern of GHA-LST accuracy; therefore, site RMSE maps under clear-sky and cloudy-sky conditions are illustrated in Figure 6.

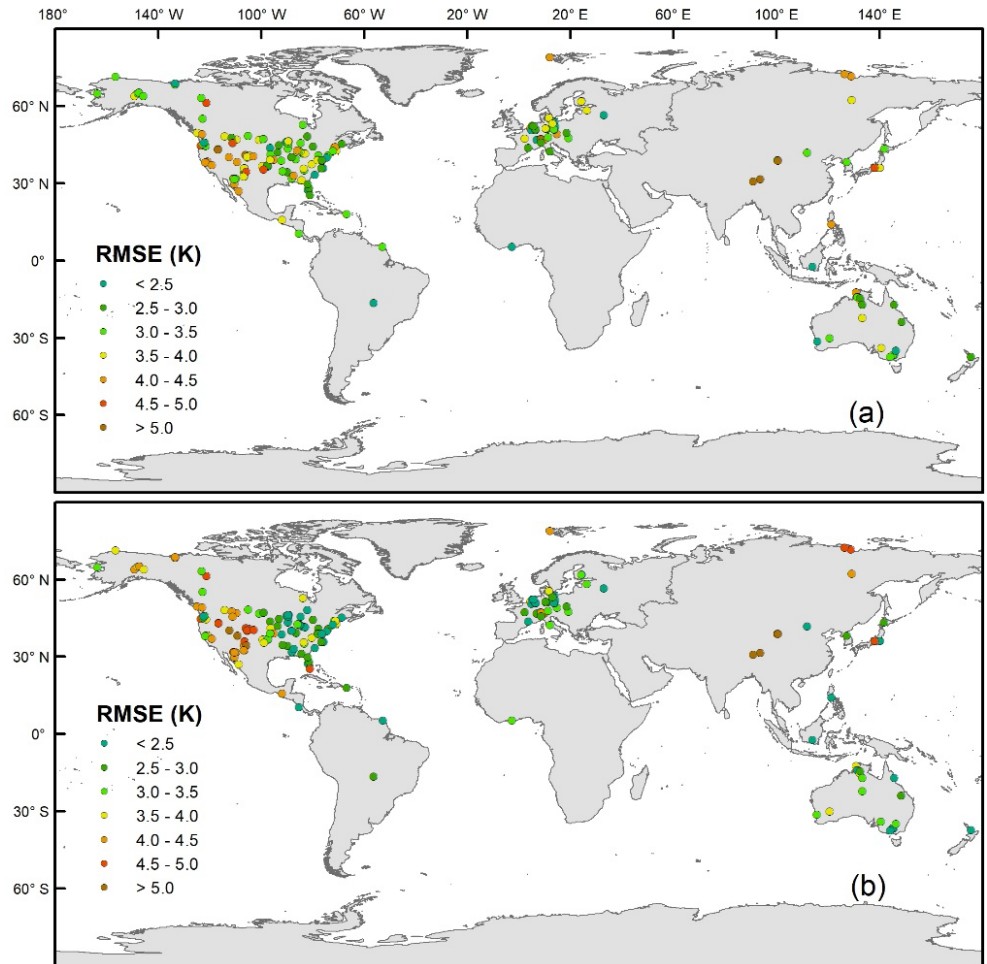

**Figure 6. Global RMSE statistics at individual sites under (a) clear-sky and (b) cloudy-sky conditions.**

The GHA-LST had similar accuracy patterns under both clear-sky and cloudy-sky conditions, and it had lower RMSEs (< 3.5 K) in eastern America, Europe, and Australia but variable RMSEs from 3.5 to 5 K in western America and the TP. The median site RMSE under clear-sky conditions was 3.18 K, with a standard deviation of 0.81 K, and the median site RMSE under cloudy-sky conditions was 2.97 K, with a standard deviation of 1.01 K, indicating that cloudy-sky results had a slightly larger spatial variance of accuracy. Validation statistics for each site are listed in the Appendix.

### 3.3 Temporal and spatial analysis

To evaluate the temporal continuity of the proposed GHA-LST dataset, the temporal LST variations from GHA-LST, CGLS, and corresponding ground measurements were compared at hourly and daily mean scales. In Figure 7, four global sites are shown as the representative sites from SURFRAD (SXF, 43.74° N, -96.62° W), BSRN (CAB, 51.97° N, 4.93° E), Fluxnet (AU-Rig, -36.65° S, 145.58° E), and AmeriFlux (US-Ro1, 44.71° N, -93.09° W). The study period was randomly chosen for different years.

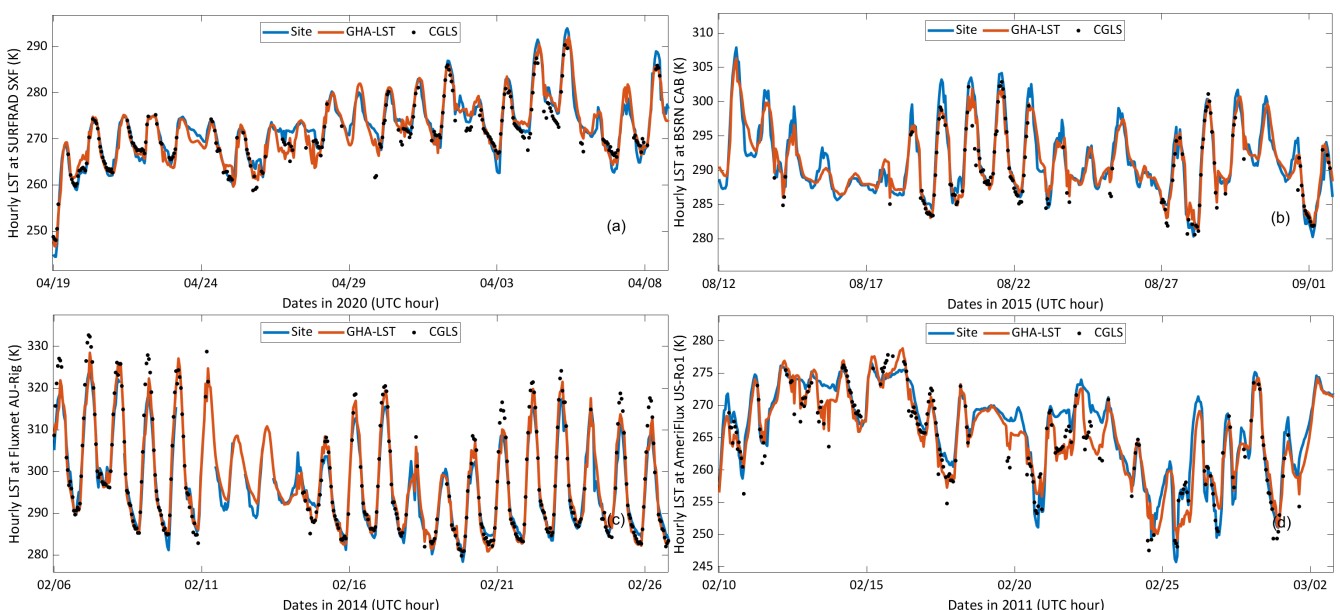

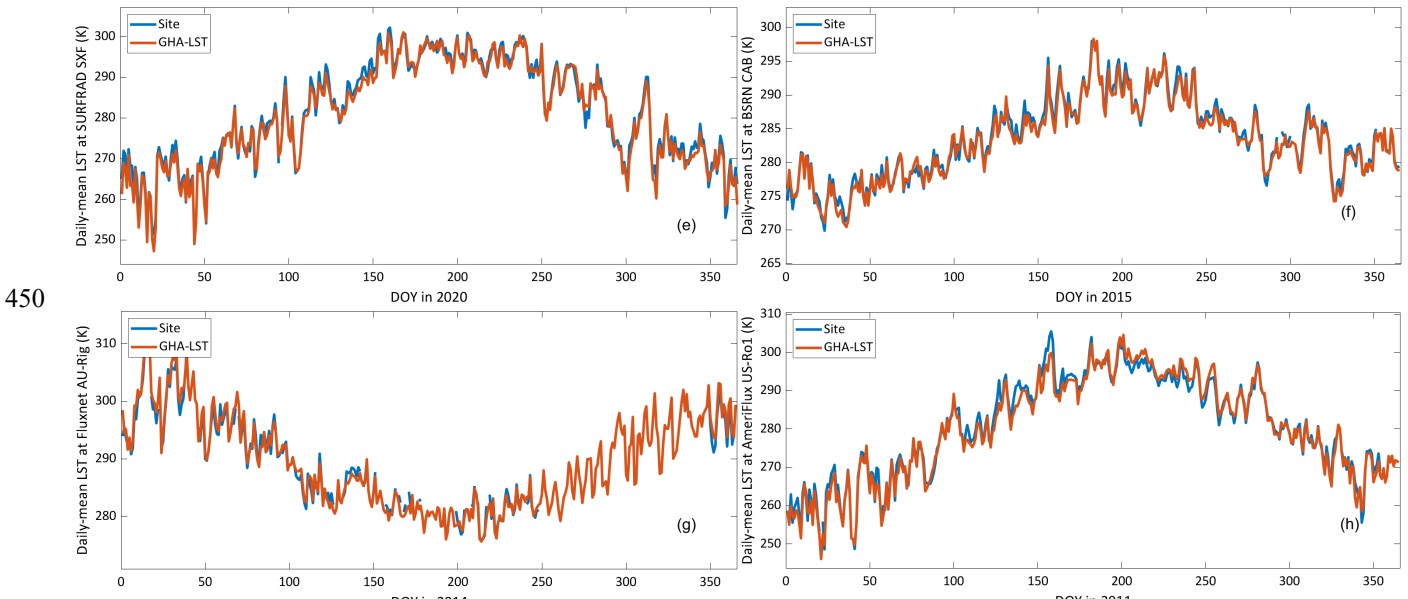

**Figure 7. Temporal variation of (a - d) hourly LST and (e - h) daily mean LST at four sites: (a, e) SURFRAD SXF, (b, f) BSRN CAB, (c, g) Fluxnet AU-Rig, and (d, h) AmeriFlux US-Ro1. Abbreviations: SURFRAD SXF, Surface Radiation Budget Sioux Falls; BSRN CAB, Baseline Surface Radiation Network Cabauw; AU-Rig, Australia- Riggs Creek; US-Ro1, United States-Ro1;**
**GHAT, global hourly, 5 km, all-sky land surface temperature; CGLS, Copernicus Global Land Service.**

The GHA-LST data have satisfactory temporal continuity and correspond to the ground measurements at hourly and daily mean scales. At the hourly scale, the hourly LST exhibits harmonic diurnal variations under clear-sky conditions, especially at Fluxnet AU-Rig (Figure 7c), where the climate is dry and cloud cover is low. In comparison, a more complicated temporal
pattern of LSTs is observed for continuous cloudy time (e.g., BSRN CAB, Figure 7b), indicating that the harmonic function-based DTC models may not work well in these cases. The GHA-LST data can capture the DTCs under both clear-sky and cloudy-sky conditions and correspond to the ground measurements and clear-sky CGLS. Certain clear-sky CGLS points are scattered and have a clear negative bias (Figure 7a) because they were detected as partially covered pixels; thus, they were not used in the data assimilation. At AmeriFlux US-Ro1 (Figure 7d), GHA-LST is more consistent with CGLS than the
ground measurements on clear-sky days; thus, we infer that US-Ro1 has a larger heterogeneity issue than the other sites.

After temporal aggregation, the daily mean LST variation in different years also demonstrated the continuity and stable accuracy of the GHA-LST. The relatively larger differences between the satellite datasets and ground measurements at noon (hourly scale) and during summer (daily mean scale) can be explained by site representativeness. A temporal variation
analysis of accuracy (Jia et al., 2022a) suggests that ground measurements generally have the lowest representativeness at noon and the RMSE statistics of hourly LST products can increase by more than 1 K from nighttime to noon. As solar radiation increases in the morning, LST has distinct warming responses over different land cover types in a pixel; thus, the spatial heterogeneity of the pixel is enhanced during daytime.

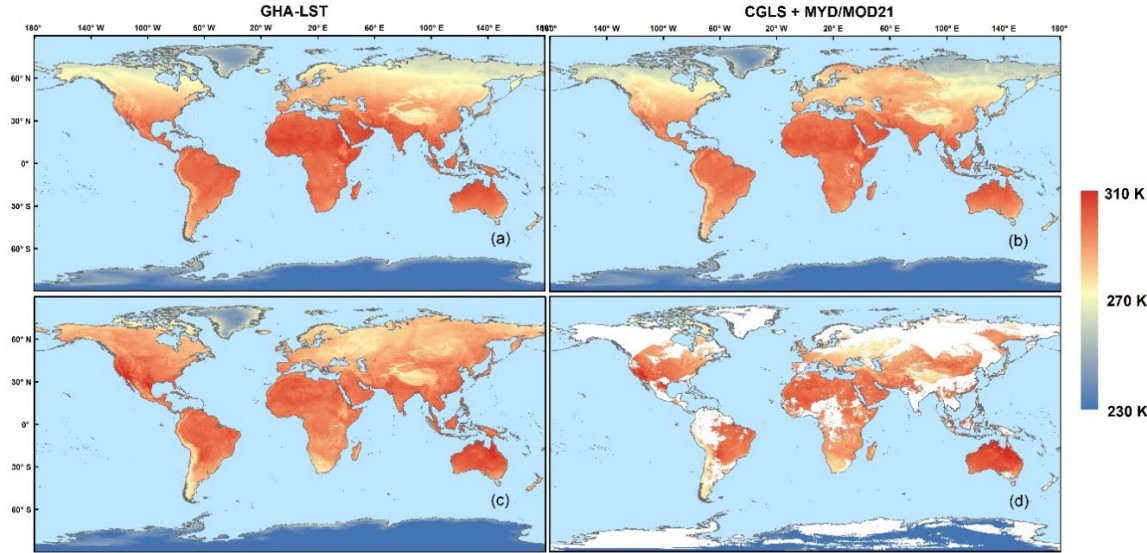

**Figure 8. Global LST maps of (a) GHA-LST annual mean in 2021, (b) CGLS + MYD/MOD21 annual mean in 2021; hourly LST maps at 02:00 UTC on September 07, 2021, of (c) GHA-LST and (d) CGLS + MYD/MOD21. Abbreviations: GHA-LST, global hourly, 5 km, all-sky land surface temperature; CGLS, Copernicus Global Land Service.**

A global all-sky LST map was analyzed to demonstrate the spatial continuity of the proposed GHA-LST dataset. The global annual mean maps of all-sky LST from GHA-LST and clear-sky LST from CGLS+MYD/MOD21 were compared (Figure 8). Overall, the GHA-LST data exhibit high spatial continuity across the globe at different time scales. The annual mean GHA-LST for 2021 (Figure 8a) illustrates a reasonable spatial pattern. The annual LSTs from CGLS+MYD/MOD21 (Figure 8b) present a systematic clear-sky bias (Ermida et al., 2019), especially in the connection regions of the CGLS and MODIS data (southwestern Canada and Siberia), where clear artificial lines are shown. In comparison, assimilating the clear-sky results to the time-evolving model can produce more spatially consistent LST maps.

At the hourly scale, the GHA-LST map (Figure 8c) can also produce the reasonable spatial variation in LST. Compared with the clear-sky pixels in Figure 8d, the cloudy-sky pixels of the GHA-LST data (Figure 8c) were well recovered. It should be noted that the clear-sky map (Figure 8b) had more spatial textures than the all-sky annual mean map (Figure 8a) because clear-sky LSTs have higher spatial heterogeneity due to solar heating. Furthermore, the various numbers of available clear-sky days in different locations may cause spurious spatial textures (e.g., lines at the connection region of CGLS and MODIS). Additionally, the GHA-LST spatiotemporally filtered clear-sky satellite LSTs using the simulated model series, which may sacrifice spatial textures for data fusion.

To evaluate the ability of GHA-LST to capture spatial textures at regional scales, the GHA-LST annual means in the Alaska and TP regions are shown in Figure 9. These two areas were selected because they are hot spot regions in terms of their

response to climate change (Kuang and Jiao, 2016; Melvin et al., 2017), and GHA-LST was recovered in these two regions from MODIS and CGLS, respectively. The corresponding annual mean skin temperatures of ERA5 and the Global Land Data Assimilation System (GLDAS) were also included for comparison. ERA5 and GLDAS were employed because global hourly all-sky LST is currently only available from reanalysis datasets and both are widely used in the relevant research (Muñoz-Sabater et al., 2021; Rodell et al., 2004). GHA-LST has a spatial pattern similar to that of the two reanalysis datasets but produces many more spatial details. GHA-LST has a spatial resolution of ~5 km; therefore, it can provide more spatial texture information than the ERA5 (0.1°) and GLDAS (0.25°) data. GLDAS data have invalid pixels, mainly because it ignores all inland lakes.

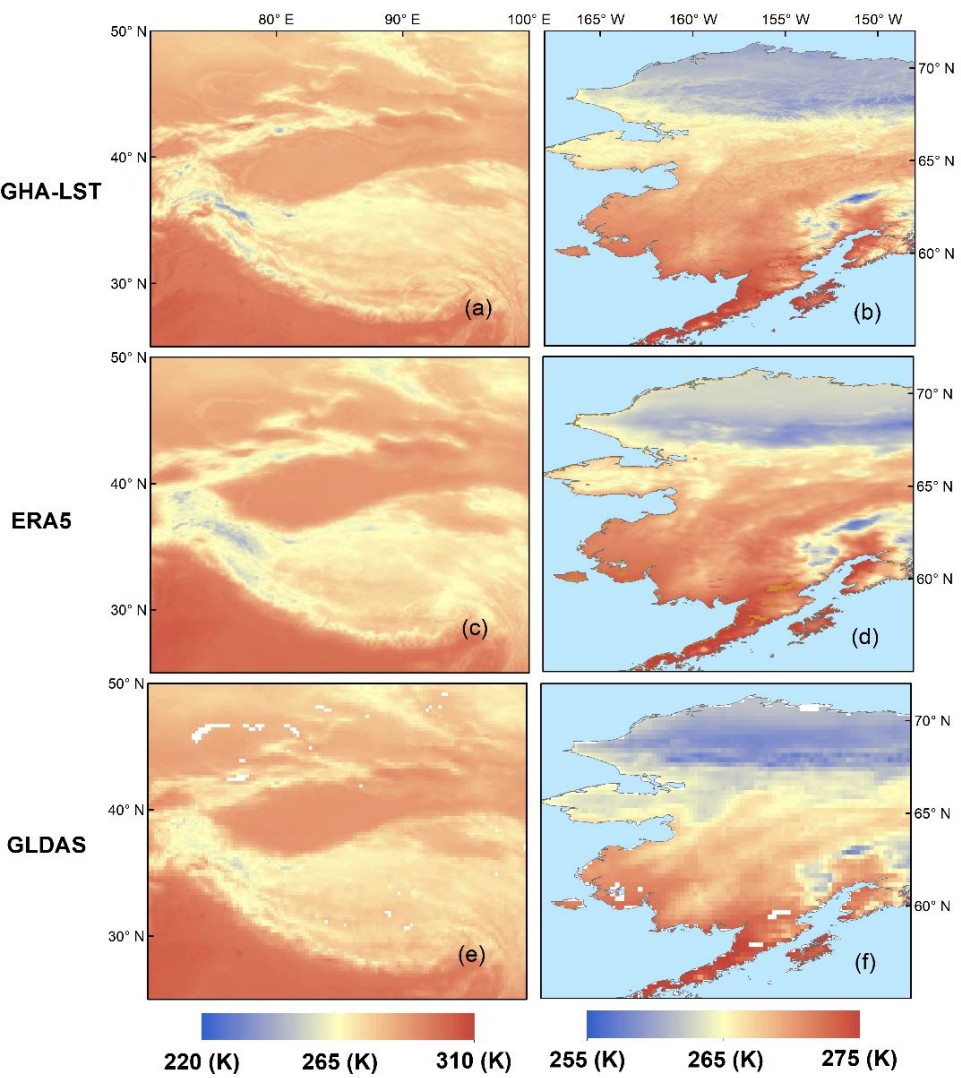

**Figure 9. Temporally averaged all-sky LST maps in 2021 from (a, b) GHA-LST, (c, d) ERA5, and (e, f) GLDAS (a, c, e) over the Tibetan Plateau and (b, d, f) Alaska. Abbreviations: GHA-LST, global hourly, 5 km, all-sky land surface temperature; GLDAS, Global Land Data Assimilation System.**

## 3.4 Global anomaly analysis

To justify the potential use of LST in climate warming-related issues, the relationship between LST and surface air temperature (AT) has recently been discussed. However, previous studies have either focused on local site scales (Hachem et al., 2012; Good, 2016; Sohrabinia et al., 2015; Mutiibwa et al., 2015) or ignored the clear-sky LST bias (Good et al., 2017). In comparison, the proposed GHA-LST provides an exceptional opportunity to spatiotemporally upscale the all-sky hourly LST to allow for comparisons with AT climate datasets. Monthly anomaly variations in global LST and AT are shown in Figure 10 by removing the seasonal cycle in the daily mean (Tmean), daily minimum temperature (Tmin), daily maximum temperature (Tmax), and diurnal temperature range (DTR).

Berkeley Earth Surface Temperatures (BEST, Rohde et al. (2013)), NASA Goddard Institute for Space Studies Surface Temperature Analysis version 4 (GISTEMP v4, Lenssen et al. (2019)), and Climatic Research Unit Temperature, version 4 (CRUTEM4, Osborn and Jones (2014)) were used to characterize the AT variation (Figure 10a). Other all-sky LST datasets were averaged and shown to verify the GHA-LST anomaly, including two MODIS-derived gap-free results, as shown in Table 1 (Hong et al., 2022; Zhang et al., 2022), and the ERA5-Land reanalysis skin temperature. Only BEST and GHA-LST can provide Tmax and Tmin; thus, they were used in Figure 10 b-d. The reference time period was 2015-2017, and the uncertainty shadow in Figure 10a is the standard deviation of the averaged LST/AT datasets.

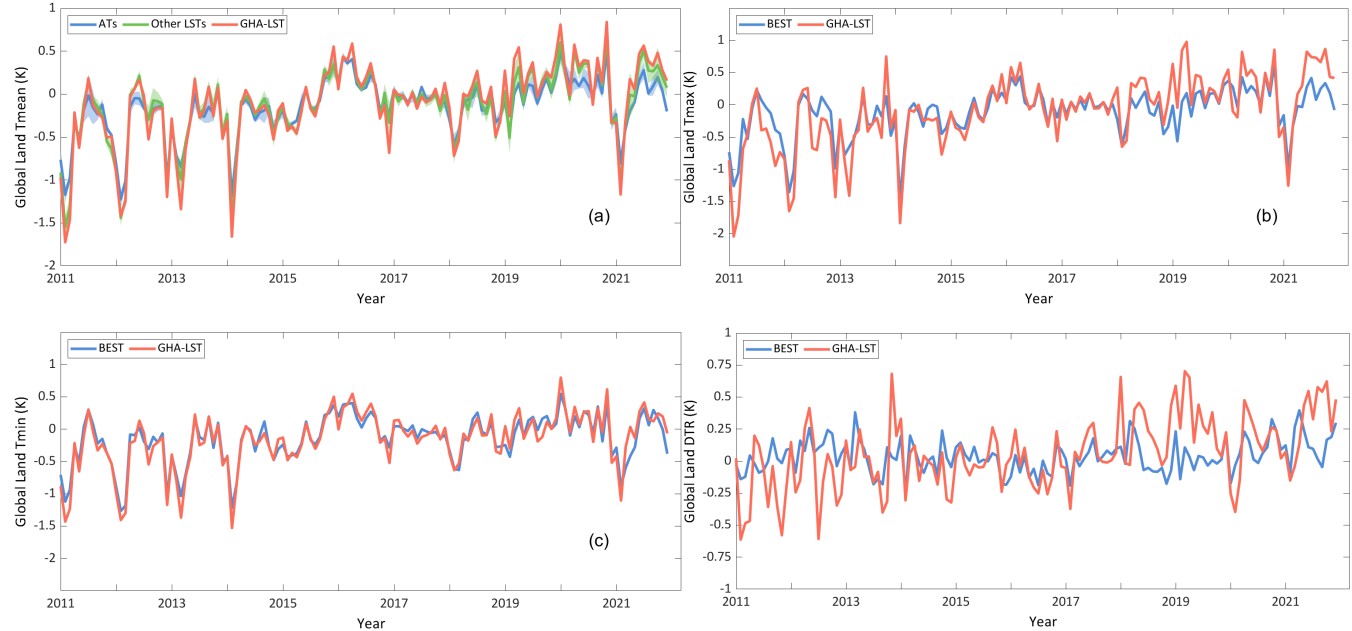

**Figure 10. Monthly anomaly variation of globally geographical weighted LST and land surface air temperature (AT) at different scales: (a) mean temperature (Tmean), (b) maximum temperature (Tmax), (c) minimum temperature (Tmin), and (d) diurnal temperature range (DTR). Abbreviations: GHA-LST, global hourly, 5 km, all-sky land surface temperature; BEST, Berkeley Earth Surface Temperatures.**

The LST anomaly couples well with the global AT anomaly at the Tmean and Tmin scales, and LST has a slightly larger amplitude than AT, whereas the Tmax and DTR of the two variables can only match the anomalous direction and the magnitude is quite different. At the Tmean scale, anomalies of GHA-LST and other LSTs have very similar variations with the AT datasets, even though they have completely different data sources (Figure 10a). These findings are consistent with the trend comparison between the ERA5-Land skin temperature and AT (Wang et al., 2022). Both datasets have limited

uncertainty (shadowed areas), indicating that they can accurately characterize the land surface thermal dynamics. In comparison, LST had a slightly larger anomaly amplitude than AT, which was mainly caused by the daytime LST. Solar heating increases the temperature difference between LSTs and ATs over different surface land cover types (Good et al., 2017). Accordingly, Tmax exhibited the largest difference (Figure 10b), especially in years with large anomalies, thus indicating that LST Tmax had a stronger response to heat anomalies. Tmin showed a higher correlation between the two

variables (Figure 10c). LST has a considerably stronger DTR disturbance than AT owing to the difference in Tmax (Figure 10d). We did not quantify the trend magnitude because the time span was only 11 years and the overall trend was affected by the value in one specific year. This analysis demonstrates the potential usefulness of GHA-LST in climate studies and global hourly AT estimates.

**4 Discussion**

The site RMSE statistics were compared with the corresponding site elevations and latitudes to detect the potential factors that impact the accuracy of the results (Figure 11).

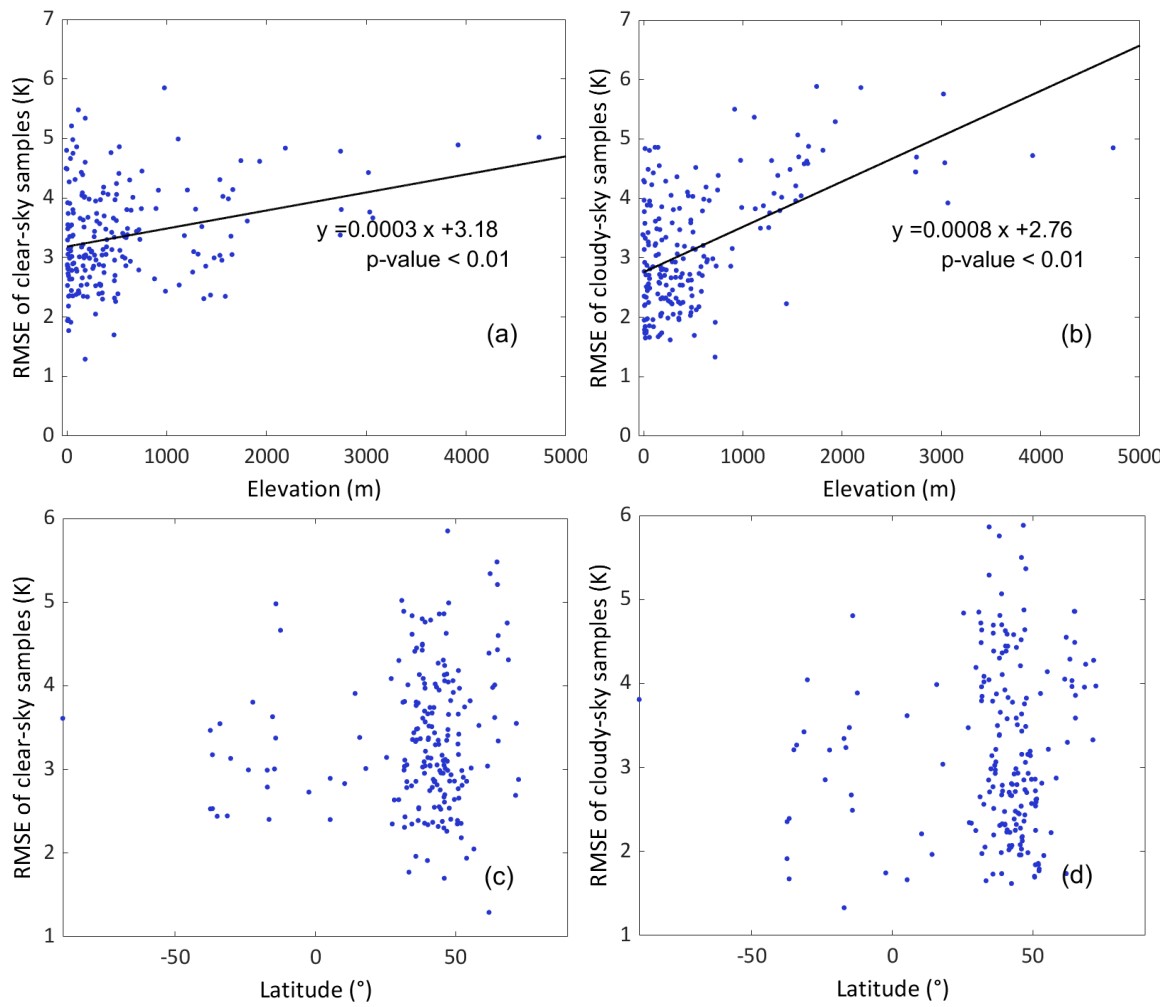

**Figure 11. Scatter plots between the site RMSE and (a, b) elevation and (c, d) latitude under (a, c) clear-sky and (b, d) cloudy-sky conditions. The significant linear relationship (p-values < 0.01) is drawn (a, b).**

The RMSE statistics at each site were mainly affected by the site elevation (Figure 11). The scatterplots of site RMSE and site elevation under clear-sky and cloud-sky conditions (Figures 7a and b) show that the linear relationship was statistically significant (p-value < 0.01). We suppose that increasing elevation will decrease the spatial representativeness of the sites; therefore, the RMSEs of the clear-sky results showed an increasing trend. In addition, the cloudy-sky results had a larger slope (Figure 11b), indicating that they were more sensitive to elevation variations. Thus, we inferred that elevation was an essential factor affecting LST recovery. In these regions with high elevation, clear-sky LSTs with larger RMSEs were assimilated in the time-evolving model, thus affecting the cloudy-sky results. In addition, modeled temperature series include higher uncertainty in these regions, and a relevant 'cool bias' issue in highlands was found in regional simulation models and global reanalysis datasets (Jia et al., 2022b; Meng et al., 2018). Although the relationship between the site RMSE and site

latitude was not statistically significant (Figure 11c and d), the GHA-LST data at higher latitudes produced higher RMSEs than the data at lower latitudes, especially under cloudy-sky conditions; thus, we inferred that high latitudes were frequently covered by clouds and fewer clear-sky LSTs could be used in the data assimilation. In addition, sites at higher latitudes are usually located in coastal areas (Figure 6a), which may limit their spatial representativeness at the 5 km scale.

Furthermore, the spatial continuity at regional scale was evaluated. A detailed mapping examination suggests that no artificial textures occurred under most conditions at mid- and low latitudes; however, at high latitudes where MODIS swath data are the basic input data, swath edges are observed on the map in some cases (Figure 12).

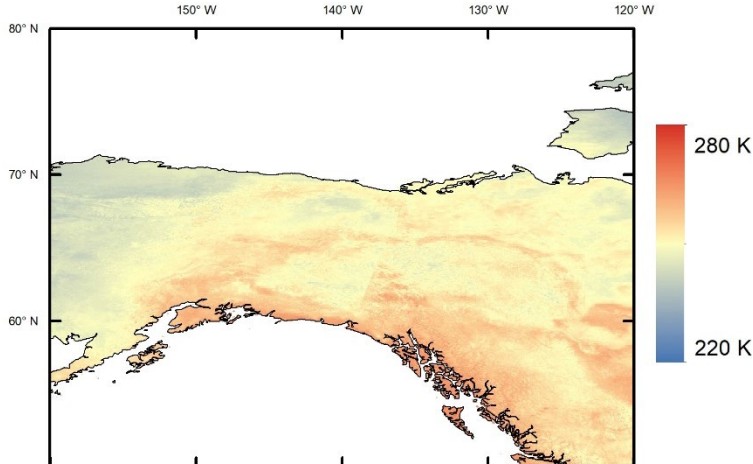

**Figure 12. Case showing the artificial texture (middle of the image) at Northwestern Canada at 05:00 UTC on January 02, 2012.**

Such spatial discontinuity occurs when the clear-sky LST retrievals within a swath have considerable temperature differences compared to that of spatially neighboring pixels that are not covered by the swath. Regions at high latitude experience longer cloud durations than those at lower latitudes; thus, pixels that are not covered by the swath might accumulate high uncertainties compared to the adjacent clear-sky retrievals. Therefore, an artificial texture remained after the data assimilation. Based on the literature review, spatial continuity is also a major issue for current MODIS LST products due to substantially different view zenith angles and view times of neighboring swaths after reprojection (Figures 14 and 17 in Li et al. (2022b)). To address such discontinuity issues, machine learning methods that incorporate additional variables for estimating cloudy-sky LST can be used in the future (Zhao et al., 2019).

**5 Data Availability**

The GHA-LST dataset from 2011 to 2021 is freely available at https://doi.org/10.5281/zenodo.7487284 (Jia et al., 2022c), and the full dataset is also accessible from glass.umd.edu/allsky_LST/GHA-LST. Quality Check (QC) flags are also

included: the Bit 0 indicates the sky condition mask (1: clear-sky, 0: cloudy-sky), and the Bit 1 is the cloud duration that represents the uncertainty level (0: <=10 days, 1: >10 days). The algorithm produced a stable accuracy within 10 days as indicated by Jia et al. (2022a).

**6 Conclusions**

LST is an essential driving factor in the surface radiation budget and hydrological cycling, and TIR-based satellite retrieval
is the primary method used to obtain LST globally. However, TIR-derived LST has numerous data gaps, mainly due to cloud cover, which seriously restricts the application of current LST products. Studies have focused on producing all-sky LST products; nevertheless, global all-sky LSTs on an hourly scale are still unavailable. Considering the high temporal variability of LST and the importance of the DTC in ET estimation, drought detection, and heatwave monitoring, we produced a global hourly, 5 km, all-sky land surface temperature dataset (GHA-LST) from 2011 to 2021. The data are recovered using CGLS
LST products from geostationary satellites and the MYD/MOD21 products from polar-orbiting satellites. Moreover, GHA-LST represents the first global gap-free LST product at an hourly scale, and it has been comprehensively validated by in situ measurements at 201 global sites in this study.

Based on the ground measurements from the SURFRAD, BSRN, Fluxnet, AmeriFlux, HRB, and TP networks, the overall
RMSE of GHA-LST is 3.31 K, with a bias of -0.57 K and $R^2$ of 0.95. The comparisons of individual accuracy suggest that the proposed GHA-LST dataset has better accuracy than the CGLS and MYD21C1 data under both clear-sky and cloudy-sky conditions. In addition, the accuracy is stable under both clear-sky and cloudy-sky conditions (RMSEs = ~3.3 K with few biases based on different sampling groups). The overall sampling amount was more than 5 million, and we suppose that the overall validation can represent the general accuracy of GHA-LST globally. In addition, after temporal aggregation to the
daily mean scale, the GHA-LST dataset produced an RMSE of 1.76 K and significantly improved the accuracy and data availability of the global daily mean LST. The individual site validation indicated that the GHA-LST dataset has similar accuracy in terms of spatial patterns under different sky conditions. In comparison, the cloudy-sky results had a larger spatial variance in accuracy.

Temporal analyses were performed for four representative sites, and the GHA-LST dataset had a high temporal continuity and was consistent with the ground measurements at hourly and daily mean scales. The temporal variation in hourly LST also illustrated that mathematically predictable DTCs cannot be obtained for locations with continuous cloudy days, thus highlighting the advantage of the time-evolving model-based method used for the GHA-LST product. Spatial analysis suggested that the GHA-LST dataset has satisfactory spatial continuity over clear-sky and cloudy-sky regions, and artificial
textures were not observed. Regional mapping analysis of the TP and Alaska regions demonstrated that GHA-LST can capture greater spatial details than reanalysis datasets, which were the only data source for obtaining hourly gap-free LSTs

before this study. The monthly anomaly analysis indicated that the GHA-LST anomalies are consistent with global AT datasets and other LST datasets at the Tmean and Tmin scales, whereas the Tmax and DTR of the LSTs and ATs are only consistent with the anomalous direction when the magnitudes are different.

In the future, additional clear-sky LST products, such as the visible infrared imaging radiometer suite (VIIRS) and advanced very high-resolution radiometer (AVHRR), can also be assimilated to increase the time span and spatial resolution of the proposed dataset. Machine learning can be employed to effectively incorporate information from ground measurements, spatial textures, and related factors (e.g., elevation, soil moisture, land cover, and wind speed). GHA-LST represents the first

gap-free LST dataset at an hourly, 5 km scale over the globe, and it has satisfactory accuracy and great potential for use in estimating global plant water stress, monitoring extreme weather, and advancing meteorological forecasting models.

## Appendix

**Table: Validation statistics of each site.**

| Network | Site | Clear-sky Samples | | | | Cloudy-sky Samples | | | |
|---------|------|------|------|-------|------|------|------|-------|------|
| | | bias | RMSE | $R^2$ | N | bias | RMSE | $R^2$ | N |
| SURFRAD | BND | -0.34 | 2.34 | 0.97 | 36983 | 0.90 | 2.32 | 0.96 | 50266 |
| | FPK | -1.17 | 2.94 | 0.97 | 36910 | -0.17 | 2.93 | 0.96 | 50501 |
| | GWN | 0.64 | 2.80 | 0.95 | 42136 | 0.64 | 2.85 | 0.91 | 44621 |
| | PSU | -0.81 | 2.75 | 0.96 | 34317 | -0.50 | 2.23 | 0.96 | 52809 |
| | SXF | -0.42 | 2.32 | 0.98 | 37649 | 0.39 | 2.48 | 0.96 | 49400 |
| BSRN | bar | 0.11 | 2.69 | 0.97 | 25941 | -0.12 | 3.33 | 0.94 | 48341 |
| | bud | 0.87 | 3.66 | 0.94 | 4942 | -0.92 | 3.49 | 0.91 | 7939 |
| | cab | -0.53 | 2.18 | 0.95 | 22129 | 0.28 | 1.84 | 0.93 | 40650 |
| | dar | -2.38 | 4.66 | 0.86 | 14516 | -1.18 | 3.89 | 0.48 | 20617 |
| | e13 | -0.31 | 2.39 | 0.97 | 37930 | 0.93 | 2.93 | 0.94 | 35193 |
| | pay | 0.79 | 2.26 | 0.96 | 29785 | -0.42 | 2.36 | 0.92 | 54290 |
| | sel | -1.55 | 3.38 | 0.90 | 3605 | -2.95 | 3.99 | 0.84 | 4843 |

| | | | | | | | | |
|---|---|---|---|---|---|---|---|---|
| | spo | 0.45 | 3.61 | 0.92 | 2681 | 0.22 | 3.81 | 0.91 | 5810 |
| | tat | 0.01 | 3.35 | 0.95 | 36718 | 0.03 | 2.51 | 0.93 | 50949 |
| | tik | 0.05 | 3.55 | 0.97 | 12411 | -0.65 | 4.28 | 0.94 | 57405 |
| | tor | -1.13 | 3.53 | 0.95 | 12559 | -0.80 | 2.87 | 0.93 | 70797 |
| Fluxnet | AU-ASM | 1.59 | 3.80 | 0.94 | 21781 | 0.61 | 3.21 | 0.90 | 12760 |
| | AU-Cpr | 0.20 | 3.55 | 0.96 | 19103 | 1.74 | 3.27 | 0.93 | 14607 |
| | AU-DaP | 3.36 | 4.98 | 0.89 | 11696 | 3.95 | 4.81 | 0.80 | 11099 |
| | AU-DaS | 0.59 | 3.37 | 0.89 | 17051 | 1.07 | 2.49 | 0.85 | 16929 |
| | AU-Dry | -1.96 | 3.63 | 0.92 | 14357 | -2.27 | 3.48 | 0.86 | 11939 |
| | AU-Emr | 0.94 | 2.99 | 0.95 | 13603 | 0.95 | 2.85 | 0.89 | 8054 |
| | AU-Gin | 0.50 | 2.44 | 0.95 | 15637 | 0.45 | 3.42 | 0.87 | 9008 |
| | AU-GWW | 0.71 | 3.13 | 0.98 | 6328 | 1.70 | 4.04 | 0.92 | 3712 |
| | AU-RDF | 0.23 | 3.01 | 0.91 | 7611 | 0.17 | 2.67 | 0.83 | 6871 |
| | AU-Rig | -0.01 | 2.53 | 0.96 | 17709 | 0.28 | 2.39 | 0.92 | 12438 |
| | AU-Rob | -0.15 | 2.79 | 0.73 | 4762 | 0.54 | 1.33 | 0.88 | 3864 |
| | AU-Stp | 0.92 | 2.99 | 0.94 | 18748 | 0.50 | 3.34 | 0.84 | 14097 |
| | AU-Whr | 0.50 | 3.17 | 0.94 | 14734 | 0.25 | 1.67 | 0.95 | 12076 |
| | AU-Wom | 0.35 | 3.47 | 0.89 | 16156 | 1.04 | 1.91 | 0.93 | 17560 |
| | AU-Ync | 0.64 | 2.44 | 0.97 | 10896 | 1.26 | 3.21 | 0.92 | 7989 |
| | BE-Lon | -0.60 | 2.35 | 0.94 | 9418 | 0.55 | 1.70 | 0.95 | 24109 |
| | CA-Gro | -0.86 | 2.40 | 0.98 | 10517 | 0.85 | 1.99 | 0.98 | 19025 |
| | CH-Cha | 2.15 | 3.48 | 0.94 | 5909 | -0.41 | 2.73 | 0.92 | 11457 |
| | CH-Dav | -0.63 | 4.14 | 0.90 | 4289 | -0.55 | 4.88 | 0.77 | 4412 |
| | CH-Fru | 3.50 | 5.85 | 0.75 | 368 | 3.21 | 4.64 | 0.85 | 90 |
| | CN-Sw2 | 1.31 | 2.37 | 0.97 | 36 | -0.29 | 2.23 | 0.46 | 11 |
| | CZ-BK1 | 0.02 | 2.64 | 0.94 | 9109 | 1.16 | 2.86 | 0.91 | 20696 |

| | | | | | | | | |
|---|---|---|---|---|---|---|---|---|
| CZ-wet | 1.64 | 4.06 | 0.88 | 7983 | 0.27 | 3.19 | 0.86 | 11327 |
| DE-Akm | 0.54 | 1.94 | 0.96 | 9771 | 1.21 | 1.95 | 0.96 | 20507 |
| DE-Geb | -0.28 | 2.66 | 0.94 | 9599 | -0.18 | 2.59 | 0.92 | 25439 |
| DE-Gri | 2.31 | 4.18 | 0.92 | 9857 | 0.94 | 2.54 | 0.93 | 24950 |
| DE-Hai | -0.77 | 3.42 | 0.89 | 4910 | 0.80 | 2.62 | 0.90 | 12100 |
| DE-Kli | -0.96 | 3.66 | 0.89 | 9558 | 0.18 | 2.72 | 0.90 | 24733 |
| DE-Lnf | -0.71 | 3.97 | 0.84 | 4835 | 0.74 | 2.63 | 0.89 | 11927 |
| DE-Obe | -1.56 | 3.30 | 0.93 | 10758 | 0.60 | 2.86 | 0.89 | 23689 |
| DE-RuR | 0.02 | 2.39 | 0.93 | 7775 | 0.17 | 1.69 | 0.95 | 22683 |
| DE-RuS | 0.22 | 2.77 | 0.92 | 5938 | 0.30 | 1.84 | 0.94 | 16662 |
| DE-SfN | 0.70 | 2.86 | 0.94 | 5497 | -1.18 | 2.70 | 0.92 | 15626 |
| DE-Spw | -0.50 | 2.35 | 0.95 | 9326 | 1.01 | 1.86 | 0.97 | 23628 |
| DE-Tha | -1.68 | 3.18 | 0.94 | 9873 | 0.20 | 2.25 | 0.93 | 25184 |
| DE-Zrk | -1.33 | 2.86 | 0.91 | 4559 | 0.00 | 1.95 | 0.93 | 9525 |
| DK-Sor | -0.94 | 3.02 | 0.91 | 7224 | 1.09 | 3.22 | 0.84 | 17380 |
| FI-Hyy | -0.65 | 1.29 | 0.99 | 24 | -0.38 | 1.74 | 0.97 | 264 |
| FR-Pue | 1.41 | 3.35 | 0.92 | 14666 | 0.24 | 2.25 | 0.92 | 17435 |
| GF-Guy | -0.16 | 2.89 | 0.33 | 7675 | 0.95 | 1.66 | 0.41 | 9819 |
| GH-Ank | 0.50 | 2.40 | 0.57 | 9314 | 2.86 | 3.62 | 0.49 | 8107 |
| IT-CA1 | -0.61 | 2.55 | 0.96 | 12906 | -1.64 | 2.81 | 0.92 | 11233 |
| IT-Isp | -0.17 | 2.67 | 0.93 | 7082 | -1.10 | 2.54 | 0.91 | 10436 |
| IT-Ren | 2.31 | 4.63 | 0.87 | 9756 | 2.95 | 5.88 | 0.71 | 12239 |
| IT-SR2 | 3.21 | 4.27 | 0.91 | 3492 | 1.56 | 2.71 | 0.90 | 17518 |
| NL-Hor | 0.55 | 2.81 | 0.86 | 2969 | 0.32 | 1.79 | 0.92 | 5737 |
| NL-Loo | -0.69 | 2.99 | 0.90 | 9372 | 0.76 | 1.77 | 0.95 | 15435 |
| RU-Fyo | -0.46 | 2.05 | 0.96 | 238 | 0.77 | 2.22 | 0.96 | 7008 |

| | | | | | | | | |
|---|---|---|---|---|---|---|---|---|
| RU-Sam | -0.76 | 2.88 | 0.98 | 1644 | 0.81 | 3.97 | 0.95 | 9583 |
| RU-SkP | -3.05 | 5.34 | 0.97 | 864 | -1.63 | 3.30 | 0.98 | 1971 |
| US-AR1 | -1.31 | 3.49 | 0.96 | 10508 | -0.24 | 3.03 | 0.95 | 6996 |
| US-AR2 | -0.83 | 3.79 | 0.95 | 7432 | -0.11 | 2.97 | 0.96 | 5464 |
| US-CRT | -0.83 | 2.78 | 0.96 | 10347 | 0.43 | 2.07 | 0.97 | 15745 |
| US-Los | -0.08 | 2.70 | 0.97 | 3021 | 0.15 | 2.17 | 0.97 | 4542 |
| US-Me2 | 0.34 | 2.75 | 0.95 | 12973 | 0.23 | 3.51 | 0.90 | 19078 |
| US-Ne1 | 0.92 | 3.13 | 0.96 | 10294 | 2.62 | 3.83 | 0.95 | 10852 |
| US-NR1 | -1.02 | 3.66 | 0.92 | 16549 | 0.47 | 3.92 | 0.86 | 18482 |
| US-Oho | -0.40 | 3.67 | 0.95 | 10223 | 1.16 | 2.70 | 0.96 | 16047 |
| US-Prr | 1.73 | 4.60 | 0.93 | 863 | 0.11 | 3.86 | 0.94 | 3674 |
| US-SRC | -0.58 | 2.43 | 0.98 | 15668 | -0.47 | 3.85 | 0.91 | 11042 |
| US-SRG | 2.25 | 3.81 | 0.95 | 21746 | 2.24 | 4.64 | 0.89 | 12345 |
| US-Syv | -2.01 | 3.01 | 0.98 | 7748 | -0.66 | 2.18 | 0.97 | 11231 |
| US-Tw1 | 2.85 | 4.50 | 0.92 | 14013 | 2.39 | 3.39 | 0.92 | 7556 |
| US-UMB | -1.62 | 3.11 | 0.97 | 11318 | 0.48 | 2.28 | 0.97 | 20943 |
| US-Var | -1.56 | 4.09 | 0.94 | 21334 | -1.75 | 3.66 | 0.92 | 13087 |
| US-WCr | -2.47 | 4.86 | 0.95 | 220 | -1.56 | 4.52 | 0.95 | 2110 |
| US-Whs | -1.15 | 2.31 | 0.98 | 19711 | -0.19 | 3.79 | 0.92 | 14735 |
| US-Wkg | -0.98 | 3.04 | 0.97 | 20263 | -0.26 | 3.96 | 0.91 | 13703 |
| US-WPT | -2.00 | 3.51 | 0.95 | 10666 | -0.21 | 2.02 | 0.96 | 14589 |
| BR-Npw | 0.76 | 2.40 | 0.91 | 8409 | 2.01 | 3.24 | 0.72 | 12547 |
| CA-SCB | 0.86 | 3.04 | 0.96 | 18819 | 0.49 | 4.05 | 0.93 | 26292 |
| DE-Dgw | -1.49 | 3.75 | 0.87 | 6103 | -0.36 | 2.81 | 0.87 | 17270 |
| FI-Sii | -1.14 | 4.39 | 0.87 | 655 | -0.44 | 4.55 | 0.88 | 19230 |
| FR-LGt | 2.16 | 4.04 | 0.91 | 3863 | 1.50 | 3.06 | 0.90 | 6792 |

| | | | | | | | | |
|---|---|---|---|---|---|---|---|---|
| | ID-Pag | 1.29 | 2.73 | 0.61 | 1361 | 0.66 | 1.74 | 0.65 | 3877 |
| | JP-BBY | 0.25 | 3.05 | 0.95 | 10928 | 0.27 | 2.92 | 0.94 | 23698 |
| | JP-Mse | -0.27 | 3.48 | 0.94 | 3501 | 0.69 | 2.20 | 0.95 | 5199 |
| | JP-SwL | -2.00 | 4.45 | 0.84 | 3522 | -1.53 | 4.39 | 0.77 | 4334 |
| | KR-CRK | -0.87 | 3.03 | 0.95 | 16752 | -0.79 | 2.31 | 0.96 | 15019 |
| | NZ-Kop | -0.50 | 2.53 | 0.90 | 16963 | 0.72 | 2.36 | 0.88 | 18010 |
| | PH-RiF | -2.70 | 3.91 | 0.75 | 2213 | -0.35 | 1.96 | 0.72 | 6215 |
| | US-Bi1 | 2.98 | 4.49 | 0.90 | 12504 | 2.92 | 4.30 | 0.84 | 8264 |
| | US-BZB | 2.51 | 5.48 | 0.93 | 8946 | 1.45 | 4.86 | 0.92 | 23668 |
| | US-EML | -0.98 | 4.01 | 0.93 | 4492 | -0.81 | 3.96 | 0.92 | 19278 |
| | US-Ho1 | -1.28 | 2.38 | 0.98 | 23048 | 0.81 | 1.96 | 0.97 | 30691 |
| | US-HRA | -0.01 | 3.36 | 0.78 | 1582 | 0.39 | 2.71 | 0.73 | 1645 |
| | US-MRM | 1.63 | 3.74 | 0.90 | 851 | 1.28 | 3.58 | 0.91 | 8514 |
| | US-NC4 | -0.14 | 1.96 | 0.95 | 6969 | 0.29 | 1.73 | 0.94 | 7481 |
| | US-NGC | 1.05 | 5.21 | 0.45 | 1226 | -1.40 | 4.49 | 0.65 | 4598 |
| | US-Sne | 2.23 | 4.80 | 0.88 | 13559 | 1.92 | 3.38 | 0.90 | 9069 |
| | US-Uaf | 2.23 | 4.43 | 0.95 | 21801 | 1.87 | 4.86 | 0.93 | 69880 |
| AmeriFlux | CA-ARB | -0.74 | 2.91 | 0.98 | 14549 | 1.34 | 3.88 | 0.95 | 26437 |
| | CA-Ca3 | 0.45 | 2.86 | 0.98 | 22291 | 1.05 | 2.57 | 0.89 | 80737 |
| | CA-Cbo | -0.10 | 2.99 | 0.96 | 30675 | 1.55 | 2.80 | 0.96 | 53145 |
| | CA-DB2 | 1.34 | 3.92 | 0.91 | 2069 | -0.29 | 3.14 | 0.84 | 9357 |
| | CA-HPC | 2.12 | 4.75 | 0.92 | 1246 | 0.43 | 3.96 | 0.90 | 5235 |
| | CA-LP1 | -0.84 | 3.82 | 0.93 | 21152 | -0.76 | 4.14 | 0.91 | 85705 |
| | CA-SMC | 1.44 | 3.98 | 0.94 | 1584 | -0.50 | 4.29 | 0.92 | 6376 |
| | CA-TVC | 0.77 | 4.31 | 0.96 | 597 | 0.44 | 4.23 | 0.93 | 7059 |
| | CR-Fsc | -1.69 | 2.83 | 0.89 | 4375 | -1.10 | 2.21 | 0.79 | 5075 |

| | | | | | | | | |
|---|---|---|---|---|---|---|---|---|
| MX-Aog | 0.68 | 4.09 | 0.82 | 13245 | -0.40 | 3.47 | 0.76 | 9890 |
| MX-Ray | -0.81 | 4.30 | 0.82 | 1696 | -0.58 | 4.19 | 0.74 | 1369 |
| PR-xGU | -1.30 | 3.01 | 0.84 | 8677 | 0.99 | 3.04 | 0.64 | 5644 |
| US-A32 | 0.58 | 2.54 | 0.97 | 8069 | 1.23 | 3.07 | 0.94 | 7247 |
| US-A74 | 0.47 | 2.53 | 0.96 | 5684 | 1.46 | 3.21 | 0.91 | 5129 |
| US-ALQ | 0.24 | 4.24 | 0.92 | 6691 | 0.02 | 3.16 | 0.93 | 9417 |
| US-Aud | 1.54 | 2.99 | 0.98 | 3896 | 2.09 | 4.49 | 0.88 | 2342 |
| US-Br1 | -0.45 | 2.88 | 0.96 | 3189 | 0.85 | 2.67 | 0.96 | 4219 |
| US-BRG | 1.03 | 3.22 | 0.94 | 15519 | 0.98 | 2.72 | 0.94 | 21042 |
| US-CPk | -0.70 | 4.79 | 0.86 | 9156 | -0.19 | 4.44 | 0.85 | 11985 |
| US-CS1 | -0.04 | 2.55 | 0.97 | 2503 | 0.56 | 2.32 | 0.97 | 4840 |
| US-Cwt | -0.34 | 3.44 | 0.87 | 18318 | -2.15 | 2.98 | 0.93 | 21479 |
| US-DFC | -0.87 | 2.82 | 0.96 | 4238 | 0.22 | 2.39 | 0.95 | 6946 |
| US-HB2 | 0.19 | 1.77 | 0.96 | 3965 | 0.63 | 1.65 | 0.95 | 4221 |
| US-HBK | -2.43 | 3.30 | 0.97 | 8713 | -1.81 | 2.99 | 0.95 | 12407 |
| US-HWB | -1.67 | 3.04 | 0.95 | 11420 | -1.09 | 2.26 | 0.96 | 15504 |
| US-Jo1 | -1.30 | 2.85 | 0.98 | 30406 | -0.74 | 4.02 | 0.92 | 24292 |
| US-KFS | 0.12 | 3.97 | 0.91 | 25297 | 1.03 | 4.36 | 0.87 | 29666 |
| US-KM4 | 0.96 | 3.44 | 0.93 | 33623 | 1.31 | 2.66 | 0.96 | 48132 |
| US-Kon | 0.11 | 4.76 | 0.89 | 12873 | 0.84 | 4.11 | 0.89 | 14146 |
| US-MC2 | 0.91 | 3.82 | 0.84 | 1515 | -0.88 | 3.16 | 0.80 | 1551 |
| US-MH1 | 2.40 | 4.13 | 0.91 | 864 | 3.32 | 5.50 | 0.87 | 1469 |
| US-Mj1 | 0.77 | 3.10 | 0.93 | 903 | 0.91 | 3.76 | 0.85 | 818 |
| US-MOz | -1.07 | 3.09 | 0.95 | 32259 | 0.46 | 1.99 | 0.97 | 38554 |
| US-Mpj | 1.14 | 4.84 | 0.93 | 49525 | 2.19 | 5.86 | 0.87 | 36528 |
| US-MRf | 0.37 | 2.52 | 0.91 | 1220 | 1.38 | 2.93 | 0.84 | 2455 |

| | US-MSR | 0.72 | 4.99 | 0.82 | 1056 | 0.20 | 5.37 | 0.72 | 1383 |
|---|---|---|---|---|---|---|---|---|---|
| | US-MVF | 0.65 | 2.54 | 0.97 | 1237 | -0.14 | 3.83 | 0.89 | 1807 |
| | US-MWA | -1.89 | 3.74 | 0.93 | 7021 | -0.32 | 2.05 | 0.97 | 13027 |
| | US-MWF | -1.28 | 2.85 | 0.95 | 6609 | 0.01 | 1.62 | 0.98 | 12025 |
| | US-MWW | -2.13 | 4.02 | 0.91 | 7136 | -1.70 | 2.80 | 0.95 | 11146 |
| | US-NC1 | 1.91 | 2.86 | 0.96 | 8453 | 1.84 | 2.77 | 0.94 | 9058 |
| | US-ONA | -0.41 | 2.35 | 0.93 | 20058 | 0.85 | 2.34 | 0.89 | 19570 |
| | US-PFb | 0.49 | 2.61 | 0.80 | 839 | 2.72 | 3.64 | 0.84 | 846 |
| | US-PFk | -0.89 | 1.70 | 0.94 | 845 | 1.44 | 2.03 | 0.96 | 885 |
| | US-PHM | -1.02 | 3.29 | 0.90 | 11040 | -0.16 | 3.78 | 0.77 | 25556 |
| | US-Rls | 0.47 | 3.99 | 0.93 | 24093 | -0.39 | 4.58 | 0.86 | 27506 |
| | US-Ro1 | 0.08 | 2.96 | 0.96 | 18488 | 0.00 | 2.46 | 0.96 | 24116 |
| | US-Rpf | -0.48 | 3.34 | 0.95 | 2054 | 0.28 | 3.59 | 0.94 | 63663 |
| | US-Seg | -0.31 | 2.35 | 0.98 | 46399 | -0.20 | 4.04 | 0.92 | 38404 |
| | US-Skr | 1.46 | 3.14 | 0.77 | 3282 | 3.46 | 4.84 | 0.65 | 2707 |
| | US-Slt | -0.30 | 1.91 | 0.97 | 8329 | 1.59 | 2.48 | 0.96 | 9199 |
| | US-TrB | -3.27 | 4.06 | 0.93 | 2256 | -2.62 | 3.43 | 0.94 | 2715 |
| | US-Tur | -1.52 | 4.31 | 0.71 | 509 | -0.19 | 4.21 | 0.68 | 1687 |
| | US-UiA | 1.49 | 3.14 | 0.96 | 3830 | 1.01 | 2.33 | 0.96 | 4907 |
| | US-Vcm | 0.10 | 3.76 | 0.90 | 45014 | 0.64 | 4.60 | 0.80 | 34198 |
| | US-Vcs | -1.09 | 3.81 | 0.90 | 24731 | -1.23 | 4.69 | 0.79 | 18661 |
| | US-Wgr | 0.14 | 3.17 | 0.92 | 3108 | -0.35 | 2.45 | 0.90 | 4191 |
| | US-Wjs | 0.59 | 4.62 | 0.93 | 42573 | 1.16 | 5.29 | 0.88 | 34057 |
| | US-Wpp | 1.26 | 4.86 | 0.77 | 2723 | 0.82 | 4.43 | 0.70 | 3381 |
| | US-Wrc | -0.80 | 3.05 | 0.92 | 13709 | -1.57 | 2.54 | 0.93 | 25363 |
| | US-xAB | -0.32 | 2.30 | 0.94 | 9821 | -0.19 | 1.96 | 0.91 | 18766 |

| | | | | | | | | |
|---|---|---|---|---|---|---|---|---|
| US-xAE | -2.71 | 4.41 | 0.94 | 15088 | -0.99 | 3.14 | 0.94 | 13825 |
| US-xBL | -1.88 | 3.04 | 0.96 | 12366 | -1.01 | 2.33 | 0.96 | 17532 |
| US-xBR | -3.18 | 3.95 | 0.97 | 14553 | -2.78 | 3.59 | 0.96 | 20114 |
| US-xCP | -2.19 | 3.05 | 0.98 | 15172 | -0.96 | 4.59 | 0.90 | 18990 |
| US-xDC | -2.22 | 3.41 | 0.97 | 12281 | -1.30 | 2.74 | 0.97 | 16593 |
| US-xDJ | -0.72 | 3.62 | 0.94 | 1186 | -1.59 | 4.03 | 0.94 | 32892 |
| US-xDL | -0.55 | 2.78 | 0.91 | 14189 | 1.05 | 2.56 | 0.92 | 17607 |
| US-xDS | -1.30 | 2.64 | 0.94 | 16311 | -0.71 | 2.34 | 0.88 | 17426 |
| US-xGR | -1.52 | 3.33 | 0.91 | 11223 | -2.71 | 3.79 | 0.91 | 16486 |
| US-xHA | -2.01 | 2.86 | 0.97 | 13716 | -0.65 | 2.07 | 0.96 | 18138 |
| US-xJE | -1.73 | 3.80 | 0.88 | 15472 | -0.20 | 2.65 | 0.90 | 16721 |
| US-xJR | -1.19 | 3.06 | 0.97 | 17076 | -0.54 | 4.09 | 0.92 | 11774 |
| US-xKA | -2.16 | 3.57 | 0.96 | 13431 | -0.84 | 2.79 | 0.95 | 14802 |
| US-xLE | -1.53 | 3.11 | 0.92 | 11741 | 0.34 | 1.98 | 0.94 | 13226 |
| US-xMB | -2.54 | 3.61 | 0.98 | 15039 | -2.31 | 4.81 | 0.93 | 13100 |
| US-xML | -1.75 | 3.37 | 0.92 | 12540 | -0.63 | 3.50 | 0.87 | 16124 |
| US-xNG | -1.91 | 3.35 | 0.97 | 10233 | -0.15 | 2.85 | 0.96 | 16173 |
| US-xNQ | -1.90 | 3.36 | 0.97 | 13902 | -1.44 | 4.63 | 0.91 | 14718 |
| US-xRM | -1.61 | 3.38 | 0.93 | 14558 | -1.59 | 4.45 | 0.84 | 16957 |
| US-xSB | -1.61 | 2.64 | 0.94 | 15547 | -1.02 | 2.25 | 0.90 | 17909 |
| US-xSC | -1.32 | 3.70 | 0.91 | 13934 | -1.00 | 2.67 | 0.94 | 20575 |
| US-xSE | -0.91 | 2.36 | 0.96 | 14088 | -0.11 | 1.74 | 0.96 | 18191 |
| US-xSL | -1.83 | 3.52 | 0.96 | 14314 | -0.63 | 4.39 | 0.89 | 15042 |
| US-xSP | -3.52 | 4.13 | 0.96 | 16381 | -2.63 | 3.88 | 0.90 | 10012 |
| US-xST | -2.19 | 2.95 | 0.98 | 10986 | -0.55 | 2.08 | 0.97 | 17499 |
| US-xTA | -3.32 | 4.01 | 0.94 | 13180 | -0.40 | 2.05 | 0.94 | 15434 |

| | | | | | | | | |
|---|---|---|---|---|---|---|---|---|
| | US-xUN | -2.53 | 3.12 | 0.98 | 13573 | -1.09 | 2.13 | 0.98 | 20054 |
| | US-xWD | -1.76 | 2.99 | 0.98 | 11784 | -0.53 | 2.44 | 0.97 | 17020 |
| HRB | ArouCJZ | -0.26 | 4.43 | 0.93 | 3624 | -0.65 | 5.76 | 0.79 | 5111 |
| | BajitanGB | -2.72 | 4.03 | 0.98 | 3925 | -2.35 | 4.70 | 0.93 | 4812 |
| | DamanCJZ | -0.03 | 2.95 | 0.96 | 3815 | 1.62 | 5.07 | 0.89 | 4729 |
| TP | biru | -0.76 | 4.89 | 0.89 | 3451 | -0.45 | 4.72 | 0.87 | 3584 |
| | namucuo | -0.23 | 4.01 | 0.90 | 3321 | 0.12 | 4.22 | 0.90 | 3412 |

**Author Contributions**

Aolin Jia: Conceptualization, data curation, formal analysis, investigation, methodology, software, validation, visualization, writing–original draft, writing–review, and editing. Shunlin Liang: Conceptualization, resources, funding acquisition, project administration, supervision, writing, review, and editing. Dongdong Wang: project administration, resources, formal analysis, investigation, methodology, writing, review, and editing. Lei Ma, Zhihao Wang, and Shuo Xu: Data curation, writing review, and editing.

**Competing Interests**

The authors declare that they have no conflicts of interest.

**Acknowledgments**

The authors acknowledge Zenodo for publishing the dataset. We are grateful to the Copernicus Global Land Service and NASA Earthdata platforms for providing satellite LST products, the ECMWF for providing the ERA5 reanalysis data, and the CERES and GLASS teams for providing the surface radiation products. We also acknowledge SURFRAD, BSRN, Fluxnet, AmeriFlux, HiWATER, and TIPEX-III for providing the field measurements. We acknowledge the insightful suggestions from João P. A. Martins and the other anonymous referee.

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
