# Peer review of "Global hourly, 5 km, all-sky land surface temperature data from 2011 to 2021 based on integrating geostationary and polar-orbiting satellite data"

_Earth System Science Data, 2022_

## Referee Comment (RC1)

Review of the paper "Global hourly, 5 km, all-sky land surface temperature data from 2011 to 2021 based on integrating geostationary and polar-orbiting satellite data"

**General Comment:**

This manuscript describes a new all-sky Land Surface Temperature dataset that extends the regional all-sky LST developed by Jia, A. et al (2022). The new dataset provides global hourly coverage based on CGLS and MODIS LSTs. The improvement with respect to both these products is clear and was well explored, both in terms of validation statistics and spatial coverage. The methodology is reasonably well explained, although some details about the inputs of the scheme (i.e., radiative fluxes and remaining inputs of the SEB; the usage of CGLS LST uncertainty in the assimilation step) and remaining limitations (e.g., its use in NRT) could be more detailed for the sake of completeness. The manuscript is very well written and structured. I would like to congratulate the authors for the great work, both in terms of the development of the method itself and the extraordinary effort that was put into its validation, which unfortunately hasn't been very common in manuscripts about this particular subject. However, I found some inconsistencies when playing with the dataset itself, which I think should be addressed and explained before the manuscript is published. They are related to data format, metadata and unexplained spatial inconsistencies.

Therefore, I recommend the publication of this manuscript, after some major issues are addressed, as detailed below.

**Specific comments:**

L53 - Jiang et al., 2015 reference missing in bibliography

L75 – ERA5-Land is missing in this list (and is cited below in L485): Muñoz-Sabater, J., Dutra, E., Agustí-Panareda, A., Albergel, C., Arduini, G., Balsamo, G., ... & Thépaut, J. N. (2021). ERA5-Land: A state-of-the-art global reanalysis dataset for land applications. Earth System Science Data, 13(9), 4349-4383.

**L259 – The meaning of $\delta$ is not explained**

L269-271 – If I understood correctly this is a limitation for the implementation of this method in NRT (as data up to d+15 would be needed). This should be stated somehow.

L287 – This is incorrect. CGLS LST comes with an uncertainty estimate per pixel which could be used in this context. It accounts for uncertainties due to GSW regression model error (which generally increases towards higher viewing angles and moister atmospheres), sensor noise, surface emissivity and total column water vapour. Uncertainties are generally below 3 K, so you could be giving too little weight to the retrieval.

You should detail a bit better how you introduce the shortwave and longwave products into the model. Please provide more details about the CERES products and other SEB inputs for clarity. For instance, I would suggest including details on the retrieval frequency, spatial resolution and limitations / uncertainties of those products, and a few sentences briefly describing the algorithms. These are critical inputs for the surface energy balance model, so it is important to highlight their strengths and weaknesses. Could you please provide a description of the validation statistics that were used? It's not clear if some kind of cloud filtering was used (I am guessing none was used). Please use the statistics recommended by the CEOS-LPV Land Surface Temperature Product Validation Best Practice Protocol https://lpvs.gsfc.nasa.gov/PDF/CEOS\_LST\_PROTOCOL\_Feb2018\_v1.1.0\_light.pdf, or discuss why you decided not to use them.

L353 - Using a nearest neighbour here introduces some additional error into the GHA-LST statistics, especially in times of day where the heating rate is higher. A more suitable way to minimize that is to interpolate LST values from consecutive hourly observations/ estimates to the MODIS acquisition time.

Fig5 – please homogenize the x and y scales in the plots

In section 3.2 I would have like to see some more information extracted from the large number of in situ sites used for validation, for instance the results discriminated by land cover type or even Koppen-Geiger climate zones. Its's also useful to have access to the individual in situ statistics, which could be provided in an annex.

(page 21) Regarding the issue of spatial consistency, as required by the Review Guidelines of the Journal, I've randomly selected a few timeslots to inspect the data.

• In the case of the GHA-LST\_2018230\_UTC09.tif, I found some strange vertical stripes and some changes in the detail of the spatial texture over Alaska:

• And a zoom over Central Asia, for the same timelot:

---

## Author Comment (AC1)

January 03, 2023

Dr. Hanqin Tian

Handling Chief Editor

Dr. Hao Shi

Handling Topical Editor

*Earth System Science Data*

Dear Dr. Tian and Dr. Shi,

Thank you so much for overseeing the review of our manuscript. We would like to thank Dr. Martins and the anonymous referee for their positive and constructive review comments. We have provided responses to their comments below and made a number of major changes in the manuscript based on their suggestion.

First, we considered the clear-sky retrieval errors in the data assimilation procedure obtained from the official LST product files. The clear-sky retrievals with large view zenith angles ($> 40°$) were not used in the new version of the proposed dataset. Second, a quality check (QC) layer was added based on the suggestions of the two reviewers, and it provides the sky condition mask and uncertainty information. Third, the outliers and incorrect patterns were resolved after revising the code.

Furthermore, statements regarding the input data and methodology were also improved based on the suggestions from Dr. Martins. A Discussion section was added to examine the current data issues based on the suggestion from the anonymous referee. The product has been reproduced and the assessment results were updated accordingly in the revised manuscript.

Thank you very much for your time and effort in editing and reviewing the manuscript. The dataset and manuscript have greatly benefited from these insightful suggestions. We look forward to working with you to move this manuscript closer to publication in *Earth System Science Data*.

Sincerely,

Aolin Jia and co-authors

University of Maryland

aolin@terpmail.umd.edu

==============================Dr. João P. A. Martins====================================

This manuscript describes a new all-sky Land Surface Temperature dataset that extends the regional all-sky LST developed by Jia, A. et al (2022). The new dataset provides global hourly coverage based on CGLS and MODIS LSTs. The improvement with respect to both these products is clear and was well explored, both in terms of validation statistics and spatial coverage. The methodology is reasonably well explained, although some details about the inputs of the scheme (i.e., radiative fluxes and remaining inputs of the SEB; the usage of CGLS LST uncertainty in the assimilation step) and remaining limitations (e.g., its use in NRT) could be more detailed for the sake of completeness. The manuscript is very well written and structured. I would like to congratulate the authors for the great work, both in terms of the development of the method itself and the extraordinary effort that was put into its validation, which unfortunately hasn't been very common in manuscripts about this particular subject. However, I found some inconsistencies when playing with the dataset itself, which I think should be addressed and explained before the manuscript is published. They are related to data format, metadata and unexplained spatial inconsistencies.

Therefore, I recommend the publication of this manuscript, after some major issues are addressed, as detailed below.

**Response**: We sincerely appreciate your support.

Specific comments:

(1) L53 - Jiang et al., 2015 reference missing in bibliography

**Response**: The reference has been added to the bibliography.

(2) L75 – ERA5-Land is missing in this list (and is cited below in L485): Muñoz-Sabater, J., Dutra, E., Agustí-Panareda, A., Albergel, C., Arduini, G., Balsamo, G., ... & Thépaut, J. N. (2021). ERA5-Land: A state-of-the-art global reanalysis dataset for land applications. Earth System Science Data, 13(9), 4349-4383.

**Response**: Thank you for your suggestion. We do not include the skin temperatures from all reanalysis datasets in Table 1 because we would like to focus on summarizing satellite-derived all-sky LST products only. This clarification has been included in the manuscript (L64-67):

> "Currently available gap-free satellite-derived LST products are summarized in Table 1. Some gap-free LST datasets are not listed in the table, such as skin temperature from reanalysis datasets (e.g., ERA5 and MERRA2) (Molod et al., 2015; Muñoz-Sabater et al., 2021), and the results of Coccia et al. (2015), as they assumed that the surface broadband emissivity was equal to 1."

In addition, an additional all-sky LST product that was released recently (Rains et al., 2022) has been added.

**Response**: Thank you for this comment. The information has been added as follows: "$\delta = 0.01$ avoids a null denominator" (L287).

(4) L269-271 – If I understood correctly this is a limitation for the implementation of this method in NRT (as data up to d+15 would be needed). This should be stated somehow.

**Response**: Thank you for your suggestion. We have provided such a statement based on the following context (L297-300):

> "It should be noted that some input data (e.g., CERES and reanalysis data) are not available at near real time (NRT); moreover, this 'likely cloud contamination' detection method also requires a 30-day time window for high-quality clear-sky LST selection, which means that the proposed cloudy-sky LST recovery method cannot be used for NRT all-sky LST production."

(5) L287 – This is incorrect. CGLS LST comes with an uncertainty estimate per pixel which could be used in this context. It accounts for uncertainties due to GSW regression model error (which generally increases towards higher viewing angles and moister atmospheres), sensor noise, surface emissivity and total column water vapour. Uncertainties are generally below 3 K, so you could be giving too little weight to the retrieval.

**Response**: Thank you for pointing out this issue. We initially tried to simplify this model without considering this information; however, in the revised version, the clear-sky retrieval error has been included to adjust the weight of clear-sky LSTs in the data assimilation (L317-319):

> "$R$ is the squared retrieval uncertainty at each pixel of CGLS and MODIS, and it is calculated based on the 'ERRORBAR_LST' or 'LST_err' data in the files."

Based on the updated assessment results, the overall validation statistics did not change significantly.

(6) You should detail a bit better how you introduce the shortwave and longwave products into the model. Please provide more details about the CERES products and other SEB inputs for clarity. For instance, I would suggest including details on the retrieval frequency, spatial resolution and limitations / uncertainties of those products, and a few sentences briefly describing the algorithms. These are critical inputs for the surface energy balance model, so it is important to highlight their strengths and weaknesses.

Response: Thank you for your suggestion. The CERES surface radiation products are hourly products that are matched with LST retrievals. Additional meta information is listed in Table 2 in the manuscript:

| Product | Variable | Temporal Resolution (°) | Spatial Resolution (°) | Usage |
|---|---|---|---|---|
| CGLS | clear-sky LST | hourly | 0.045 | LST for recovery |
| Swath MOD/MYD21 | clear-sky LST | instantaneous | 1 km | LST for recovery |
| ERA5 | clear-sky DLW & ULW | hourly | 0.25 | time-evolving model |
| GLASS | BBE | 8-day | 0.05 | time-evolving model |
| CERES | all-sky and clear-sky DSR and DLW | hourly | 1 | cloud effect |
| GLASS | surface albedo | 8-day | 0.05 | cloud effect |
| GLASS | LAI | daily | 0.05 | cloud effect |

An introduction to the CERES products has also been added (L164-174):

> "Global hourly surface DLW and downward shortwave radiation (DSR) from CERES satellite products were used to estimate the cloud effect. To monitor cloud radiative forcing, the CERES project retrieved global, gap-free, hourly DSR and DLW in both all-sky (realistic) and theoretically cloud-free conditions (Doelling et al., 2016). CERES utilized the same strategy as this study to generate global hourly radiation products by combining remote sensing observations from multiple GEO sensors and two MODIS sensors. The CERES surface shortwave radiation and longwave radiation (Doelling et al., 2013) were estimated based on the Langley Fu–Liou radiative transfer theory (Fu et al., 1997), the cloud properties were obtained from microwave cloud products (Minnis et al., 2020), and the aerosol optical depth was based on the MODIS aerosol product (Remer et al., 2006). Surface CERES downward radiation fluxes have an overall bias (standard deviation) of 3.0 W m$^{-2}$ (5.7%) for shortwave and $-4.0$ W m$^{-2}$ (2.9%) for longwave radiation, which have been validated based on 85 sites (Rutan et al., 2015). CERES has been extensively evaluated and is generally considered a benchmark for satellite radiation products for assessments and inter-comparisons (Jia et al., 2018; Li et al., 2022; Wang and Dickinson, 2013)."

Moreover, a discussion regarding the strengths and weaknesses of utilizing CERES surface radiation products and additional details describing how radiation products help the SEB scheme were added in the subsequent paragraph of the manuscript.

Strengths (L175-177):

"To improve the production efficiency, the complicated downward longwave parameterization schemes in Jia et al. (2022) were replaced by directly exploiting the CERES dataset and converting its cloud radiative forcing into the corresponding cloud cooling/warming effect."

Additional introduction regarding how the radiation products help the SEB scheme for cloud effect estimation are provided as follows (L177-180):

"Specifically, the CERES DSR difference between all-sky and clear-sky schemes was considered cloud DSR forcing, and combined with the GLASS surface albedo data, the cloud net shortwave forcing was computed. Cloud DLR forcing represents the difference between CERES all-sky and clear-sky DLR products, and the corresponding net longwave forcing was estimated using an optimization method (Section 2.2.4).

Weaknesses (L181-186):

"In addition, according to previous studies (Wang and Dickinson, 2013; Zhang et al., 2015), the impact of the coarse spatial resolution of CERES downward radiation can be ignored because it has less heterogeneity than surface variables. CERES products were bilinearly interpolated to match the spatial scale of the CGLS LST. However, this assumption may introduce a certainty degree of uncertainty in areas with rugged terrain because complicated terrain in a coarse pixel may still affect the downward radiation components and increase the heterogeneity."

(7) Could you please provide a description of the validation statistics that were used? It's not clear if some kind of cloud filtering was used (I am guessing none was used). Please use the statistics recommended by the CEOS-LPV Land Surface Temperature Product Validation Best Practice Protocol https://lpvs.gsfc.nasa.gov/PDF/CEOS_LST_PROTOCOL_Feb2018_v1.1.0_light.pdf , or discuss why you decided not to use them.

**Response**: Additional information and an explanation have been added to the manuscript (L211-217) as follows:

"The following validation metrics are used in this study: N is the sample amount; bias, also called mean bias error (MBE), represents the systematic errors/differences between LST products and ground measurements; root-mean-square-error (RMSE) characterizes the actual uncertainty caused by bias and random error; and $R^2$ indicates the overall goodness of fit based on a 1:1 line. These metrics are commonly used for LST validation. The standard deviation (SD) of the differences between LST products and site measurements was not used because it provides similar information as the RMSE but cannot reflect errors caused by systematic bias; thus, the SD is generally smaller than the RMSE."

In the updated validation statistics in the revised manuscript, the samples were filtered by the 'likely cloud contamination detection' method used with the data assimilation method, as indicated in response (4). Moreover, the MYD21C1 samples (not the MODIS swath LST), which were not used in the production process and were only included for accuracy comparisons, were selected based on the QC flag. They passed the cloud contamination detection test based on the work of Ma et al. (2020) as stated in Section 3.6 (L395-396).

(8) L353 - Using a nearest neighbour here introduces some additional error into the GHA-LST statistics, especially in times of day where the heating rate is higher. A more suitable way to minimize that is to interpolate LST values from consecutive hourly observations/ estimates to the MODIS acquisition time.

**Response**: We followed your suggestion and revised the text (L382-384):

> "Because MYD21C1 may have a slightly different observation time ($< 0.5$ h) relative to GHA-LST, it was converted to the nearest UTC time based on the diurnal cycle recorded by site observations to match the GHA-LST observation time."

It should be noted that initially we did not directly use MYD21 samples to validate our all-sky results because MYD21 is a clear-sky only product. The data from MYD21 and this study were validated by corresponding site samples. Therefore, we converted the MYD21 samples using corresponding DTC information from site measurements to match with the GHA-LST recording time.

(9) Fig5 – please homogenize the x and y scales in the plots

**Response**: Thank you for pointing out this issue. It has been fixed in the revised manuscript.

(10) In section 3.2 I would have like to see some more information extracted from the large number of in situ sites used for validation, for instance the results discriminated by land cover type or even Koppen-Geiger climate zones. Its's also useful to have access to the individual in situ statistics, which could be provided in an annex.

**Response:** Thank you for providing such a valuable suggestion. An assessment has been added in Table 3 (L420) and the statistics for each site are in the Appendix in the revised manuscript.

(11) (page 21) Regarding the issue of spatial consistency, as required by the Review Guidelines of the Journal, I've randomly selected a few timeslots to inspect the data.

[Figure]

• In the case of the GHA-LST_2018230_UTC09.tif, I found some strange vertical stripes and some changes in the detail of the spatial texture over Alaska:

**Response:** Thank you so much for providing such a detailed examination. We have revised this issue based on the following explanation: the "strange vertical stripes" are caused by a coding bug in the MODIS swath data gridding, and the spatial detail inconsistency is restricted after considering the error patterns of clear-sky retrievals and removing the pixels with a large view zenith angle ($> 40°$). The revised image for GHA-LST_2018230_UTC09 is as follows:

[Figure]

• And a zoom over Central Asia, for the same timelot:

This location showed a similar issue as that observed for Alaska and has been updated:

[Figure]

• This kind of features is common across other timeslots: there are areas with lot of finer scale structure and then abrupt changes to wider smoother areas. I think it should be acknowledged as a weakness of the product and the causes should be further explored. In fact, in L471, the opposite is suggested: "At the hourly scale, the GHA-LST map (Figure 9c) can also produce the reasonable spatial variation in LST without artificial textures"

**Response**: Thank you for your suggestion. In the revised manuscript, these texture issues have been addressed by considering the impact of a larger view zenith angle and retrieval errors. However, cases have been described in the Discussion section of the revised manuscript (L550-586), as requested by Reviewer #2, to demonstrate the limitations of the dataset.

• I would also recommend releasing the dataset in the NetCDF4 format, as it would reach a wider audience (in particular the climate modelling community).

**Response**: Thank you for your suggestion. The new version has now been released in NetCDF (NC) format.

• I couldn't find the way to convert from int32 to physical values (i.e., where is the scaling_factor and add_offset values we must use to produce actual LSTs from the information within the geoTiffs?).

**Response**: The scaling factor was 0.01, which was provided in a 'readme' file. The new version has been released in NC format.

• I would also like to be able to isolate clear and cloudy situations in the analysis, would the authors be able to provide a cloud mask together with the product itself as an ancillary variable?

**Response**: We followed the suggestions from both you and Reviewer #2, and the QC band has been added: the Bit-0 indicates the sky condition (1: clear sky, 0: cloudy sky), and the Bit-1 indicates the cloud duration, which represents the uncertainty level (0: <=10 days, 1: >10 days). The algorithm reaches a stable state within 10 days as indicated by Jia et al. (2022). Such information has been introduced in the product file.

• There is no uncertainty information, as required by the journal guidelines.

**Response**: The uncertainty information has been included in the revised dataset as indicated above.

L572 – I strongly disagree with this speculation; as the authors say across the manuscript, most of the lower accuracies over mountainous regions can be explained by lower site representativity; therefore MW sensors with larger footprints will not help correcting those issues. This was discussed in Martins et al (2019).

**Response**: This statement has been revised accordingly.

L577 -and maybe soil moisture?

**Response**: This statement has been revised accordingly (L629).

L750 – Please check this reference, there is a typo.

**Response**: This typo has been revised accordingly.

=======================================Reviewer #2=======================================

The authors refined the SEB-based cloudy-sky LST recovery method and produced a global 15 hourly, 5 km, all-sky land surface temperature (GHA-LST) dataset from 2011 to 2021.

Overall, the topic is interesting and relevant to ESSD. I have some major concerns as follows:

**Response**: We sincerely appreciate your support.

(1) In my opinion, data description paper not only provide data generation (methods, technological process and data availability) but also data usage (quality control, limitation, and so on). Although most similar studies didn't provide related explanation, it is very important for data description paper.

**Response**: Thank you so much for this suggestion. Based on the suggestion by both you and Dr. Martins, we have included a QC band in each file:

The first bit indicates the sky condition (1: clear sky, 0: cloudy sky), and the second bit indicates the cloud duration, which represents the uncertainty level (0: <=10 days, 1: >10 days). The algorithm reaches a stable state within 10 days as indicated by Jia et al. (2022). Such information has been introduced in the product file.

(2) The Swath MOD/MYD21 LSTs are used for recovery. As we know, LSTs can vary between the viewing zenith angle and acquisition time. The differences in LST measured at nadir and off-nadir can be up to 5–10 K. This effect must also be considered for the GHA-LST dataset.

**Response**: Thank you so much for this suggestion. Additional processes have been added to minimize the impact of large view zenith angles (VZA):

1) Considering the large data volume of MODIS LSTs at high latitudes, pixels with a VZA larger than 40° are not used in the data assimilation. The clear-sky retrievals generally have VZA < 65°, and studies have proven that pixels with VZA < 40° show stable retrieval accuracy and are seldom affected by the VZA (Guillevic et al., 2013; Li et al., 2014) (L145-146).

2) We included retrieval errors obtained from official files. The official LST products have provided larger error values for pixels that are influenced by the large VZA or emissivity uncertainty, and their corresponding weight will decrease in the data assimilation (L318-319).

(3) I had downloaded the dataset and found some outliers. Therefore, a comprehensive discussion is needed.

**Response**: Thank you so much for pointing out these mistakes. The outliers and spatial discontinuities have been revised accordingly. An example has been provided in the response (11) to Dr. Martins, and a corresponding discussion has been also provided.

A Discussion section has been included in the manuscript based on examples selected from the updated dataset. Data limitations and reasonable explanations are also provided in this section.

**References**

[revised manuscript text omitted]